# Gathering and Exploiting Higher-Order Information when Training Large Structured Models

## Abstract

When training large models, such as neural networks, the full derivatives of order 2 and beyond are usually inaccessible, due to their computational cost. This is why, among the second-order optimization methods, it is very common to bypass the computation of the Hessian by using first-order information, such as the gradient of the parameters (e.g., quasi-Newton methods) or the activations (e.g., K-FAC).

In this paper, we focus on the exact and explicit computation of projections of the Hessian and higher-order derivatives on well-chosen subspaces, which are relevant for optimization. Namely, for a given partition of the set of parameters, it is possible to compute tensors which can be seen as "higher-order derivatives according to the partition", at a reasonable cost as long as the number of subsets of the partition remains small.

Then, we propose an optimization method exploiting these tensors at order 2 and 3 with several interesting properties, including: it outputs a learning rate per subset of parameters, which can be used for hyperparameter tuning; it takes into account long-range interactions between the layers of the trained neural network, which is usually not the case in similar methods (e.g., K-FAC); the trajectory of the optimization is invariant under affine layer-wise reparameterization.

## 1 Introduction

The appealing theoretical properties of Newton's method have led to numerous attempts to adapt it to neural network optimization. Therefore, the study of the Hessian of a loss according to many parameters has become a research area in itself, leading to a large number of methods to approximate it accurately with low computational cost.

**Newton's method applied to neural network optimization.** When it comes to neural networks, Newton's method suffers from several problems. Some are *technical*, such as building an accurate and computationally efficient method for estimating the Hessian. But some of them are *essential*, in the sense that they cannot be solved by a perfect knowledge of the full Hessian alone. For instance, several works (Sagun et al., 2018) have shown that many eigenvalues of the Hessian are close to zero when training neural networks, making it impossible to use Newton's method in practice, even if the Hessian is perfectly known.

Therefore, in this work, we do not aim to build a technique to estimate the Hessian accurately and efficiently. Instead, we propose to take a step aside and focus on two related goals. First, we aim to efficiently access higher-order information in order to use it for optimization. To do this, we leverage the available computational tools, that is, automatic differentiation and partition of the set of parameters into tensors stored on a GPU. Second, we aim to build a second-order optimization method with properties similar to those of Newton's method (but inevitably weaker). In particular, we want a method that is invariant by layer-wise affine reparameterizations of the model, and that provides a Hessian-inspired matrix showing the interactions between each pair of tensors, along with layer-wise step sizes. Naturally, the computational cost should remain reasonable, and the problem of close-to-zero eigenvalues of the Hessian has to solved.

**First contribution: extracting higher-order information.** Formally, we study a loss $\mathcal{L}$ to minimize according to a vector of parameters $\boldsymbol{\theta} \in \mathbb{R}^P$, which can be represented as a tuple of tensors $(\mathbf{T}^1, \cdots, \mathbf{T}^S)$. In a multilayer perceptron with $L$ layers, the $S = 2L$ tensors $(\mathbf{T}^s)_{1 \le s \le 2L}$ are the tensors of weights and the vectors of biases of each layer. In that case, $S \ll P$. Within this framework, we propose a technique *summarizing* the order-$d$ derivative of the loss, which is a tensor belonging to $\mathbb{R}^{P^d}$, into a tensor belonging to $\mathbb{R}^{S^d}$, which is significantly smaller and easier to compute.

**Second contribution: second-order optimization method.** Then, we make use of the preceding technique at orders 2 and 3 to build a second-order optimization method. Formally, the method presented here and Newton's method look alike: in both cases, a linear system $\mathbf{H}_0 \mathbf{x} = \mathbf{g}_0$ has to be solved (according to $\mathbf{x}$), where $\mathbf{g}_0$ and $\mathbf{H}_0$ contain respectively first-order and second-order information about $\mathcal{L}$. Despite this formal resemblance, the difference is enormous: with Newton's method, $\mathbf{H}_0$ is equal to the Hessian $\mathbf{H}$ of $\mathcal{L}$ of size $P \times P$, while with ours, $\mathbf{H}_0$ is equal to a matrix $\bar{\mathbf{H}}$ of size $S \times S$. Thus, $\bar{\mathbf{H}}$ is undoubtedly smaller and easier to compute than $\mathbf{H}$ when $S \ll P$. Nevertheless, since $\bar{\mathbf{H}}$ is a dense matrix, it still contains information about the interactions between the tensors $\mathbf{T}^s$ when they are used in $\mathcal{L}$. This point is crucial because most second-order optimization methods applied to neural networks use a simplified version of the Hessian (or its inverse), usually a diagonal or block-diagonal approximation, which ignores the interactions between layers. Finally, we propose an anisotropic version of Nesterov's cubic regularization (Nesterov & Polyak, 2006), which uses order-3 information to regularize $\bar{\mathbf{H}}$ and avoid instabilities when computing $\bar{\mathbf{H}}^{-1} \bar{\mathbf{g}}$.

A proof-of-concept of this method is given in Section 5. Apart from that, this method provides a way to evaluate the interactions between layers from a training perspective, it outputs by-layer learning rates (which makes it applicable in hyperparameter tuning), and its training trajectory is invariant by layer-wise affine reparameterizations (so it preserves some interesting theoretical properties of Newton's method).

**Structure of the paper.** First, we show the context and motivation of our work in Section 2. Then, we provide in Sections 3 and 4 standalone presentations of the contributions, respectively the higher-order information extraction technique and the second-order optimization method. In Section 5, we present experimental results showing that the developed methods are usable in practice. Finally, we discuss the results in Section 6.

## 2 CONTEXT AND MOTIVATION

### 2.1 HIGHER-ORDER INFORMATION

It is not a novel idea to extract higher-order information about a loss at a minimal computational cost to improve optimization. This is typically what is done by Dangel (2023), although it does not go beyond the second-order derivative. In this line of research, the *Hessian-vector product* (Pearlmutter, 1994) is a decisive tool, that allows to compute the projection of higher-order derivatives in given directions at low cost (see App. A). For derivatives of order 3 and beyond, Nesterov's cubic regularization of Newton's method (Nesterov & Polyak, 2006) uses information of order 3 to avoid too large training steps. Incidentally, we develop an anisotropic variant of this in Section 4.2.

### 2.2 SECOND-ORDER METHODS

The Hessian $\mathbf{H}$ of the loss $\mathcal{L}$ according to the vector of parameters $\boldsymbol{\theta}$ is known to contain useful information about $\mathcal{L}$. Above all, the Hessian is used to develop second-order optimization algorithms. Let us denote by $\boldsymbol{\theta}_t$ the value of $\boldsymbol{\theta}$ at time step $t$, $\mathbf{g}_t \in \mathbb{R}^P$ the gradient of $\mathcal{L}$ at step $t$ and $\mathbf{H}_t$ its Hessian at step $t$. One of the most widely known second-order optimization method is Newton's method, whose step is (Nocedal & Wright, 1999, Chap. 3.3):

$$\boldsymbol{\theta}_{t+1} := \boldsymbol{\theta}_t - \mathbf{H}_t^{-1} \mathbf{g}_t. \tag{1}$$

Under certain conditions, including strong convexity of $\mathcal{L}$, the convergence rate of Newton's method is quadratic (Nocedal & Wright, 1999, Th. 3.7), which makes it very appealing. Besides, other methods use second-order information without requiring the full computation of the Hessian. For

instance, Cauchy's steepest descent (Cauchy, 1847) is a variation of the usual gradient descent, where the step size is tuned by extracting very little information from the Hessian:

$$\boldsymbol{\theta}_{t+1} := \boldsymbol{\theta}_t - \eta_t^* \mathbf{g}_t, \quad \text{where} \quad \eta_t^* := \frac{\mathbf{g}_t^T \mathbf{g}_t}{\mathbf{g}_t^T \mathbf{H}_t \mathbf{g}_t}, \tag{2}$$

where the value of $\mathbf{g}_t^T \mathbf{H}_t \mathbf{g}_t$ can be obtained with little computational cost (see Appendix A). However, when optimizing a quadratic function $f$ with Cauchy's steepest descent, $f(\theta_t)$ is known to decrease at a rate $(\frac{\lambda_{\max} - \lambda_{\min}}{\lambda_{\max} + \lambda_{\min}})^2$, where $\lambda_{\max}$ and $\lambda_{\min}$ are respectively the largest and the smallest eigenvalues of the Hessian of $f$ (Luenberger & Ye, 2008, Chap. 8.2, Th. 2). If the Hessian of $f$ is strongly anisotropic, then this rate is close to one and optimization is slow. For a comparison of the two methods, see (Gill et al., 1981; Luenberger & Ye, 2008; Nocedal & Wright, 1999).

Finally, there should be some space between Newton's method, which requires the full Hessian $\mathbf{H}$, and Cauchy's steepest descent, which requires minimal and computationally cheap information about $\mathbf{H}$. The optimization method presented in Section 4 explores this in-between space.

**Quasi-Newton methods.** When the parameter space is high-dimensional, computation of the Hessian $\mathbf{H}_t$ and inversion of the linear system $\mathbf{g}_t = \mathbf{H}_t \mathbf{x}$ are computationally intensive. Quasi-Newton methods are designed to avoid any direct computation of the Hessian, and make extensive use of gradients and finite difference methods to approximate the direction of $\mathbf{H}_t^{-1} \mathbf{g}_t$. For a list of quasi-Newton methods, see (Nocedal & Wright, 1999, Chap. 8). However, Nocedal & Wright (1999) argue that, since it is easy to compute the Hessian by using Automatic Differentiation (AutoDiff), quasi-Newton methods tend to lose their interest. Nevertheless, they should remain useful when such computation is too difficult.

**Applications to deep learning.** Many methods overcome the curse of the number of parameters by exploiting the structure of the neural networks. It is then common to neglect interactions between layers, leading to a (block)-diagonal approximation of the Hessian. A first attempt has been made by Wang & Lin (1998): they divide the Hessian into blocks, following the division of the network into layers, and its off-diagonal blocks are removed. From another perspective, Ollivier (2015) keeps this block-diagonal structure, but performs an additional approximation on the remaining blocks.

More recently, K-BFGS has been proposed (Goldfarb et al., 2020), which is a variation of the quasi-Newton method BFGS with block-diagonal approximation and an approximate representation of these blocks. In a similar spirit, the Natural Gradient method TNT (Ren & Goldfarb, 2021) also exploits the structure of neural networks by performing a block-diagonal approximation. Finally, AdaHessian (Yao et al., 2021) efficiently implements a second-order method by approximating the Hessian by its diagonal.

**Kronecker-Factored Approximate Curvature (K-FAC).** This approximation of the Hessian was proposed in (Martens & Grosse, 2015) in the context of neural network training. K-FAC exploits the specific architecture of neural networks to output a cheap approximation of the true Hessian. Despite its scalability, K-FAC suffers from several problems. First, the main approximation is quite rough, since "[it assumes] statistical independence between products [...] of unit activities and products [...] of unit input derivatives" (Martens & Grosse, 2015, Sec. 3.1). Second, even with an approximation of the Hessian, one has to invert it, which is computationally intensive even for small networks. To overcome this difficulty, a block-(tri)diagonal approximation of the inverse of the Hessian is made, which eliminates many of the interactions between the layers.

**Summarizing the Hessian.** Also, (Lu et al., 2018) proposes to approximate the Hessian with a matrix composed of blocks in which all coefficients are identical. Thus, the Hessian can be compressed into a smaller matrix that looks like the summary of Hessian matrix $\bar{\mathbf{H}}$ used in Section 4. In a completely different setup, Yuan et al. (2022) proposes a "Sketched Newton-Raphson", which is driven by the same spirit as the method presented in Section 4: instead of dealing with a complicated large matrix, one should "project" it onto spaces of lower dimension.

**Invariance by affine reparameterization.** Several optimization methods, such as Newton's, have an optimization step invariant by affine reparameterization of $\boldsymbol{\theta}$ (Amari, 1998) (Nesterov, 2003, Chap.

4.1.2). Specifically, when using Newton's method, it is equivalent to optimize $\mathcal{L}$ according to $\boldsymbol{\theta}$ and according to $\tilde{\boldsymbol{\theta}} = \mathbf{A}\boldsymbol{\theta} + \mathbf{B}$ ($\mathbf{A} \in \mathbb{R}^{P \times P}$ invertible, $\mathbf{B} \in \mathbb{R}^P$). This affine-invariance property holds even if the function $\mathcal{L}$ to minimize is a negative $\log$-likelihood, and one chooses to minimize $\tilde{\boldsymbol{\theta}}$ by the *natural gradient* method (Amari, 1998). This method also requires computing the Hessian of $\mathcal{L}$ at some point.

**Methods based on the moments of the gradients.** Finally, many methods acquire information about the curvature of the loss surface by using only the gradients. For instance, Shampoo (Gupta et al., 2018) uses second-moment information of the accumulated gradients.

## 2.3 MOTIVATION

**What are we really looking for?** The methods that aim to estimate the Hessian matrix $\mathbf{H}$ or its inverse $\mathbf{H}^{-1}$ in order to imitate Newton's method implicitly assume that Newton's method is adapted to the current problem. This assumption is certainly correct when the loss to optimize is strongly convex. But, when the loss is not convex and very complicated, e.g. when training a neural network, this assumption is not justified. Worse, it has been shown empirically that, at the end of the training of a neural network, the eigenvalues of the Hessian are concentrated around zero (Sagun et al., 2018), with only a few large positive eigenvalues. Therefore, Newton's method itself does not seem to be recommended for neural network training, so we may not need to compute the full Hessian at all, which would relieve us of a tedious, if not impossible, task.

To avoid such problems, it is very common to regularize the Hessian by adding a small, constant term $\lambda \mathbf{I}$ to it (Nocedal & Wright, 1999, Chap. 6.3). Also, trust-region Newton methods are designed to handle non-positive-definite Hessian matrices (Nocedal & Wright, 1999, Chap. 6.4) (Nash, 1984).

**Importance of the interactions between layers.** Also, some empirical works have shown that the role and the behavior of each layer must be considered along its interactions with the other layers, which emphasize the importance of off-diagonal blocks in the Hessian or its inverse. We give two examples. First, Zhang et al. (2022) has shown that, at the end of their training, many networks exhibit a strange feature: some (but not all) layers can be reinitialized to their initial value with little loss of the performance. Second, Kornblith et al. (2019) has compared the similarity between the representations of the data after each layer: changing the number of layers can qualitatively change the similarity matrix of the layers (Kornblith et al., 2019, Fig. 3). Among all, these results motivate our search for mathematical objects that show how layers interact.

**Per-layer scaling of the learning rates.** A whole line of research is concerned with building a well-founded method for finding a good scaling for the initialization distribution of the parameters, and for the learning rates, which can be chosen layer-wise. For instance, a layer-wise scaling for the weights was proposed and theoretically justified in the paper introducing the Neural Tangent Kernels (Jacot et al., 2018). Also, in the "feature learning" line of work, (Yang & Hu, 2021) proposes a relationship between different scalings related to weight initialization and training. Therefore, there is an interest in finding a scalable and theoretically grounded method to build per-layer learning rates.

**Unleashing the power of AutoDiff.** Nowadays, several libraries provide easy-to-use automatic differentiation packages that allow the user to compute numerically the gradient of a function, and even higher-order derivatives.[1] Ignoring the computational cost, the full Hessian could theoretically be computed numerically without any approximation. To make this computation feasible, one should aim for an simpler goal: instead of computing the Hessian, one can consider a smaller matrix, consisting of projections of the Hessian.

Moreover, one might hope that such projections would "squeeze" the close-to-zero eigenvalues of the Hessian, so that the eigenvalues of the projected matrix would not be too close to zero.

## 3 SUMMARIZING HIGHER-ORDER INFORMATION

Let us consider the minimization of a loss function $\mathcal{L} : \mathbb{R}^P \to \mathbb{R}$ according to a variable $\boldsymbol{\theta} \in \mathbb{R}^P$.

---

[1]With PyTorch: torch.autograd.grad.

**Full computation of the derivatives.** The order-$d$ derivative of $\mathcal{L}$ at a point $\boldsymbol{\theta}$, that we denote by $\frac{\mathrm{d}^d \mathcal{L}}{\mathrm{d}\boldsymbol{\theta}^d}(\boldsymbol{\theta})$, can be viewed as either a $d$-linear form (see Dieudonné (1960) and Appendix L) or as an order-$d$ tensor belonging to $\mathbb{R}^{P^d}$. For convenience, we will use the latter: the coefficients of the tensor $\mathbf{A} = \frac{\mathrm{d}^d \mathcal{L}}{\mathrm{d}\boldsymbol{\theta}^d}(\boldsymbol{\theta}) \in \mathbb{R}^{P^d}$ are $A_{i_1,\cdots,i_d} = \frac{\partial^d \mathcal{L}}{\partial \theta_{i_1} \cdots \partial \theta_{i_d}}(\boldsymbol{\theta})$, where $(i_1, \cdots, i_d) \in \{1, \cdots, P\}^d$ is a multi-index. For a tensor $\mathbf{A} \in \mathbb{R}^{P^d}$, we will use the following notation for tensor contraction:

$$\forall (\mathbf{u}_1, \cdots, \mathbf{u}_d) \in \mathbb{R}^P \times \cdots \times \mathbb{R}^P, \quad \mathbf{A}[\mathbf{u}^1, \cdots, \mathbf{u}^d] := \sum_{i_1=1}^{P} \cdots \sum_{i_d=1}^{P} A_{i_1,\cdots,i_d} u_{i_1}^1 \cdots u_{i_d}^d. \quad (3)$$

The order-$d$ derivative $\frac{\mathrm{d}^d \mathcal{L}}{\mathrm{d}\boldsymbol{\theta}^d}(\boldsymbol{\theta}) \in \mathbb{R}^{P^d}$ contains $P^d$ scalars. But, even when considering its symmetries, it is computationally too expensive to compute it exactly for $d \geq 2$ in most cases. For instance, it is not even possible to compute numerically the full Hessian of $\mathcal{L}$ according to the parameters of a small neural network, i.e., with $P = 10^5$ and $d = 2$, the Hessian contains $P^d = 10^{10}$ scalars.

**Terms of the Taylor expansion.** At the opposite, one can obtain cheap higher-order information about $\mathcal{L}$ at $\boldsymbol{\theta}$ by considering a specific direction $\mathbf{u} \in \mathbb{R}^P$. The Taylor expansion of $\mathcal{L}(\boldsymbol{\theta} + \mathbf{u})$ gives:

$$\mathcal{L}(\boldsymbol{\theta} + \mathbf{u}) = \mathcal{L}(\boldsymbol{\theta}) + \sum_{d=1}^{D} \frac{1}{d!} \frac{\mathrm{d}^d \mathcal{L}}{\mathrm{d}\boldsymbol{\theta}^d}(\boldsymbol{\theta})[\mathbf{u}, \cdots, \mathbf{u}] + o(\|\mathbf{u}\|^D). \quad (4)$$

The terms of the Taylor expansion contain higher-order information about $\mathcal{L}$ in the direction $\mathbf{u}$. In particular, they can be used to predict how $\mathcal{L}(\boldsymbol{\theta})$ would change if $\boldsymbol{\theta}$ was translated in the direction of $\mathbf{u}$. Additionally, computing the first $D$ terms has a complexity of order $D \times P$, which is manageable even for large models. The trick that allows for such a low complexity, the *Hessian-vector product*, was proposed by Pearlmutter (1994) and is recalled in Appendix A.

**An intermediate solution.** Now, let us assume that, in the practical implementation of a gradient-based method of optimization of $\mathcal{L}(\boldsymbol{\theta})$, $\boldsymbol{\theta}$ is represented by a tuple of tensors $(\mathbf{T}^1, \cdots, \mathbf{T}^S)$. So, each Taylor term can be expressed as:

$$\frac{\mathrm{d}^d \mathcal{L}}{\mathrm{d}\boldsymbol{\theta}^d}(\boldsymbol{\theta})[\mathbf{u}, \cdot\cdot, \mathbf{u}] = \sum_{s_1=1}^{S} \cdots \sum_{s_d=1}^{S} \frac{\partial^d \mathcal{L}}{\partial \mathbf{T}^{s_1} \cdots \partial \mathbf{T}^{s_d}}(\boldsymbol{\theta})[\mathbf{U}^{s_1}, \cdots, \mathbf{U}^{s_d}] = \mathbf{D}_{\boldsymbol{\theta}}^d(\mathbf{u})[\mathbb{1}_S, \cdots, \mathbb{1}_S], \quad (5)$$

where $\mathbb{1}_S \in \mathbb{R}^S$ is a vector full of ones, the tuple of tensors $(\mathbf{U}^1, \cdots, \mathbf{U}^S)$ represents $\mathbf{u}$,[2] and $\mathbf{D}_{\boldsymbol{\theta}}^d(\mathbf{u}) \in \mathbb{R}^{S^d}$ is a tensor of order $d$ with size $S$ in every dimension with values:

$$(\mathbf{D}_{\boldsymbol{\theta}}^d(\mathbf{u}))_{s_1,\cdots,s_d} = \frac{\partial^d \mathcal{L}}{\partial \mathbf{T}^{s_1} \cdots \partial \mathbf{T}^{s_d}}(\boldsymbol{\theta})[\mathbf{U}^{s_1}, \cdots, \mathbf{U}^{s_d}] \quad (6)$$

$$= \sum_{i_1=1}^{P_{s_1}} \cdots \sum_{i_1=1}^{P_{s_d}} \frac{\partial^d \mathcal{L}}{\partial T_{i_1}^{s_1} \cdots \partial T_{i_d}^{s_d}}(\boldsymbol{\theta}) U_{i_1}^{s_1} \cdots U_{i_d}^{s_d}, \quad (7)$$

where $P_s$ is the number of coefficients of the tensor $\mathbf{T}^s$. Thus, $\mathbf{D}_{\boldsymbol{\theta}}^d(\mathbf{u})$ is a tensor of order $d$ and size $S$ in every dimension resulting from a partial contraction of the full derivative $\frac{\mathrm{d}^d \mathcal{L}}{\mathrm{d}\boldsymbol{\theta}^d}(\boldsymbol{\theta})$. Moreover, the trick of Pearlmutter (1994) also applies to the computation of $\mathbf{D}_{\boldsymbol{\theta}}^d(\mathbf{u})$, which is then much less expensive to compute than the full derivative (see Appendix A).

**Properties of $\mathbf{D}_{\boldsymbol{\theta}}^d(\mathbf{u})$.** We show a comparison between the three techniques in Table 1. If $S$ is small enough, computing $\mathbf{D}_{\boldsymbol{\theta}}^d(\mathbf{u})$ becomes feasible for $d \geq 2$. For usual multilayer perceptrons with $L$ layers, there is one tensor of weights and one vector of biases per layer, so $S = 2L$. This allows to compute $\mathbf{D}_{\boldsymbol{\theta}}^d(\mathbf{u})$ in practice for $d = 2$ even when $L \approx 20$.

According to Eqn. 5, the Taylor term can be obtained by full contraction of $\mathbf{D}_{\boldsymbol{\theta}}^d(\mathbf{u})$. However, $\mathbf{D}_{\boldsymbol{\theta}}^d(\mathbf{u})$, is a tensor of size $S^d$, and cannot be obtained from the Taylor term, which is only a scalar. Thus, the tensors $\mathbf{D}_{\boldsymbol{\theta}}^d(\mathbf{u})$ extract more information than the Taylor terms, while keeping a reasonable computational cost. Moreover, their off-diagonal elements give access to information about one-to-one interactions between tensors $(\mathbf{T}^1, \cdots, \mathbf{T}^S)$ when they are processed in the function $\mathcal{L}$.

---

[2] $(\mathbf{U}^1, \cdots, \mathbf{U}^S)$ is to $\mathbf{u}$ as $(\mathbf{T}^1, \cdots, \mathbf{T}^S)$ is to $\boldsymbol{\theta}$.

Table 1: Comparison between three techniques extracting higher-order information about $\mathcal{L}$: size of the result and complexity of the computation.

| Technique | Size | Complexity |
|---|---|---|
| Full derivative $\frac{\mathrm{d}^d \mathcal{L}}{\mathrm{d}\boldsymbol{\theta}^d}(\boldsymbol{\theta})$ | $P^d$ | $P^d$ |
| Taylor term $\mathbf{D}_{\boldsymbol{\theta}}^d(\mathbf{u})[\mathbb{1}_S, \cdots, \mathbb{1}_S]$ | $1$ | $d \times P$ |
| Tensor $\mathbf{D}_{\boldsymbol{\theta}}^d(\mathbf{u})$ | $S^d$ | $S^{d-1} \times P$ |

# 4 A SCALABLE SECOND-ORDER OPTIMIZATION METHOD

## 4.1 PRESENTATION OF THE METHOD

The method presented here consists in partitioning the set of indices of parameters $\{1, \cdots, P\}$ into $S$ subsets $(\mathcal{I}_s)_{1 \leq s \leq S}$, assigning for all $1 \leq s \leq S$ the same learning rate $\eta_s$ to the parameters $(\theta_p)_{p \in \mathcal{I}_s}$, and finding the vector of learning rates $\boldsymbol{\eta} = (\eta_1, \cdots, \eta_S)$ optimizing the decrease of the loss $\mathcal{L}$ for the current training step $t$, by using its order-2 Taylor approximation.[3] Formally, given a direction $\mathbf{u}_t \in \mathbb{R}^P$ in the parameter space (typically, $\mathbf{u}_t = \mathbf{g}_t$, the gradient) and $\mathbf{U}_t := \mathrm{Diag}(\mathbf{u}_t) \in \mathbb{R}^{P \times P}$, we consider the training step: $\boldsymbol{\theta}_{t+1} := \boldsymbol{\theta}_t - \mathbf{U}_t \mathbf{I}_{P:S} \boldsymbol{\eta}_t$, that is a training step in a direction based on $\mathbf{u}_t$, distorted by a subset-wise step size $\boldsymbol{\eta}_t$.

Then, we minimize the order-2 Taylor approximation of $\mathcal{L}(\boldsymbol{\theta}_{t+1}) - \mathcal{L}(\boldsymbol{\theta}_t)$: $\boldsymbol{\Delta}_2(\boldsymbol{\eta}_t) := -\mathbf{g}_t^T \mathbf{U}_t \mathbf{I}_{P:S} \boldsymbol{\eta}_t + \frac{1}{2} \boldsymbol{\eta}_t^T \mathbf{I}_{S:P} \mathbf{U}_t \mathbf{H}_t \mathbf{U}_t \mathbf{I}_{P:S} \boldsymbol{\eta}_t$, which gives:

$$\boldsymbol{\theta}_{t+1} = \boldsymbol{\theta}_t - \mathbf{U}_t \mathbf{I}_{P:S} \boldsymbol{\eta}_t^*, \qquad \boldsymbol{\eta}_t^* := (\mathbf{I}_{S:P} \mathbf{U}_t \mathbf{H}_t \mathbf{U}_t \mathbf{I}_{P:S})^{-1} \mathbf{I}_{S:P} \mathbf{U}_t \mathbf{g}_t, \qquad (8)$$

where $\mathbf{I}_{S:P} \in \mathbb{R}^{S \times P}$ is the *partition matrix*, verifying $(\mathbf{I}_{S:P})_{sp} = 1$ if $p \in \mathcal{I}_s$ and 0 otherwise, and $\mathbf{I}_{P:S} := \mathbf{I}_{S:P}^T$. Alternatively, $\boldsymbol{\eta}_t^*$ can be written (details are provided in Appendix B):

$$\boldsymbol{\eta}_t^* = \bar{\mathbf{H}}_t^{-1} \bar{\mathbf{g}}_t, \quad \text{where:} \quad \bar{\mathbf{H}}_t := \mathbf{I}_{S:P} \mathbf{U}_t \mathbf{H}_t \mathbf{U}_t \mathbf{I}_{P:S} \in \mathbb{R}^{S \times S}, \quad \bar{\mathbf{g}}_t := \mathbf{I}_{S:P} \mathbf{U}_t \mathbf{g}_t \in \mathbb{R}^S. \quad (9)$$

With the notation of Section 3, $\bar{\mathbf{H}}_t = \mathbf{D}_{\boldsymbol{\theta}_t}^{(2)}(\mathbf{u}_t)$ and $\bar{\mathbf{g}}_t = \mathbf{D}_{\boldsymbol{\theta}_t}^{(1)}(\mathbf{u}_t)$. Incidentally, computing $\bar{\mathbf{H}}$ is of complexity $SP$, and solving the system $\bar{\mathbf{H}}\mathbf{x} = \bar{\mathbf{g}}$ is of complexity $S^2$.

## 4.2 REGULARIZING $\bar{\mathbf{H}}$ BY USING ORDER-3 INFORMATION

The method proposed in Section 4.1 requires to compute $\boldsymbol{\eta}^* = \bar{\mathbf{H}}^{-1} \bar{\mathbf{g}}$. Usually, inverting such a linear system at every step is considered as hazardous and unstable. Therefore, when using Newton's method, instead of computing descent direction $\mathbf{u} := \mathbf{H}^{-1}\mathbf{g}$, it is very common to add a regularization term: $\mathbf{u}_\lambda := (\mathbf{H} + \lambda \mathbf{I})^{-1}\mathbf{g}$ (Nocedal & Wright, 1999, Chap. 6.3).

However, the theoretical ground of such a regularization technique is not fully satisfactory. Basically, the main problem is not having a matrix $\bar{\mathbf{H}}$ with close-to-zero eigenvalues: after all, if the loss landscape is very flat in a specific direction, it is better to make a large training step. The problem lies in the order-2 approximation of the loss made in the training step 8, as well as in Newton's method: instead of optimizing the true decrease of the loss, we optimize the decrease of its order-2 approximation. Thus, the practical question is: does this approximation faithfully model the loss at the current point $\boldsymbol{\theta}_t$, in a region that also includes the next point $\boldsymbol{\theta}_{t+1}$?

To answer this question, one has to take into account order-3 information, and regularize $\bar{\mathbf{H}}$ in such a way that the resulting update remains in a region around $\boldsymbol{\theta}_t$ where the cubic term of the Taylor approximation is negligible. In practice, we propose an anisotropic version of Nesterov's cubic regularization (Nesterov & Polyak, 2006).

---

[3]With the notation of Section 3, $\mathcal{I}_s$ is the set of indices $p$ of the parameters $\theta_p$ belonging to the tensor $\mathbf{T}^s$, so the scalars $(\theta_p)_{p \in \mathcal{I}_s}$ correspond to the scalars belonging to $\mathbf{T}^s$. So, everything is as if a specific learning rate $\eta_s$ is assigned to each $\mathbf{T}^s$.

**Anisotropic Nesterov cubic regularization.** By using the technique presented in Section 3, the diagonal coefficients $(D_1, \cdots, D_S)$ of $\mathbf{D}_{\boldsymbol{\theta}}^{(3)}(\mathbf{u}) \in \mathbb{R}^{S \times S \times S}$ are available with little computational cost. Let: $\mathbf{D} := \mathrm{Diag}(|D_1|^{1/3}, \cdots, |D_S|^{1/3}) \in \mathbb{R}^S$.

We modify the method of Nesterov & Polyak (2006) by integrating an anisotropic factor $\mathbf{D}$ into the cubic term. Thus, our goal is to minimize according to $\boldsymbol{\eta}$ the function $T$: $T(\boldsymbol{\eta}) := -\boldsymbol{\eta}^T \bar{\mathbf{g}} + \frac{1}{2} \boldsymbol{\eta} \bar{\mathbf{H}} \boldsymbol{\eta} + \frac{\lambda_{\mathrm{int}}}{6} \|\mathbf{D} \boldsymbol{\eta}\|^3$, where $\lambda_{\mathrm{int}}$ is the *internal damping* coefficient, which can be used to tune the strength of the cubic regularization. Under conditions detailed in Appendix D, this minimization problem is equivalent to finding a solution $\boldsymbol{\eta}_*$ such that:

$$\boldsymbol{\eta}_* = \left( \bar{\mathbf{H}} + \frac{\lambda_{\mathrm{int}}}{2} \|\mathbf{D} \boldsymbol{\eta}_*\| \mathbf{D}^2 \right)^{-1} \bar{\mathbf{g}}, \tag{10}$$

which is no more than a regularized version of 8. Finally, this multi-dimensional minimization problem boils down to a scalar root finding problem (see Appendix D).

### 4.3 PROPERTIES

The final method is a combination of the training step 8 with regularization 10:

**Method 1.** *Training step* $\boldsymbol{\theta}_{t+1} = \boldsymbol{\theta}_t - \mathbf{U}_t \mathbf{I}_{P:S} \boldsymbol{\eta}_t^*$, *where* $\boldsymbol{\eta}_t^*$ *is the solution with the largest norm* $\|\mathbf{D}_t \boldsymbol{\eta}\|$ *of the equation:* $\boldsymbol{\eta} = \left( \bar{\mathbf{H}}_t + \frac{\lambda_{\mathrm{int}}}{2} \|\mathbf{D}_t \boldsymbol{\eta}\| \mathbf{D}_t^2 \right)^{-1} \bar{\mathbf{g}}_t$.

**Encompassing Newton's method and Cauchy's steepest descent.** Without the cubic regularization ($\lambda_{\mathrm{int}} = 0$), Newton's method is recovered when using the *discrete partition*, that is, $S = P$ with $\mathcal{I}_s = \{s\}$ for all $s$, and Cauchy's steepest descent is recovered when using the *trivial partition*, that is, $S = 1$ with $\mathcal{I}_1 = \{1, \cdots, P\}$. See Appendix C for more details.

**No need to compute or approximate the full Hessian.** The full computation of the Hessian $\mathbf{H}_t \in \mathbb{R}^{P \times P}$ is not required. Instead, one only needs to compute the $S \times S$ matrix $\bar{\mathbf{H}}_t := \mathbf{I}_{S:P} \mathbf{U}_t \mathbf{H}_t \mathbf{U}_t \mathbf{I}_{P:S}$, which can be done efficiently by computing $\mathbf{u}^T \mathbf{H}_t \mathbf{v}$ for a number $S \times S$ of pairs of well-chosen directions $(\mathbf{u}, \mathbf{v}) \in \mathbb{R}^P \times \mathbb{R}^P$. This property is especially useful when $S \ll P$. When optimizing a neural network with $L = 10$ layers and $P = 10^6$ parameters, one can naturally partition the set of parameters into $S = 2L$ subsets, each one containing either all the weights or all the biases of each of the $L$ layers. In this situation, one has to solve a linear system of size $2L = 20$ at each step, which is much more reasonable than solving a linear system of $P = 10^6$ equations. We call this natural partition of the parameters of a neural network the *canonical partition*.

**No need to solve a large linear system.** Using Equations 8 or 10 requires solving only a linear system of $S$ equations, instead of $P$ in Newton's method. With the cubic regularization, only a constant term is added to the complexity, since it is a matter of scalar root finding.

**The interactions between different tensors are not neglected.** The matrix $\bar{\mathbf{H}}_t$, which simulates the Hessian $\mathbf{H}_t$, is basically dense: it does not exhibit a (block-)diagonal structure. So, the interactions between subsets of parameters are taken into account when performing optimization steps. In the context of neural networks with the canonical partition, this means that interactions between layers are taken into account during optimization, even if the layers are far from each other. This is a major advantage over many existing approximations of the Hessian or its inverse, which are diagonal or block-diagonal.

**Invariance by subset-wise affine reparameterization.** As showed in Appendix E, under a condition on the directions $\mathbf{u}_t$,[4] the trajectory of optimization of a model trained by Method 1 is invariant by affine reparameterization of the sub-vectors of parameters $\boldsymbol{\theta}_{\mathcal{I}_s} := \mathrm{vec}(\{\theta_p : p \in \mathcal{I}_s\})$. Let $(\alpha_s)_{1 \le s \le S}$ and $(\beta_s)_{1 \le s \le S}$ be a sequence of nonzero scalings and a sequence of offsets, and $\tilde{\boldsymbol{\theta}}$ such that, for all $1 \le s \le S$, $\tilde{\boldsymbol{\theta}}_{\mathcal{I}_s} = \alpha_s \boldsymbol{\theta}_{\mathcal{I}_s} + \beta_s$. Then, the training trajectory of the model is the same with both parameterizations $\boldsymbol{\theta}$ and $\tilde{\boldsymbol{\theta}}$. This property is desirable in the case of neural networks, where

---

[4]This holds typically if $\mathbf{u}_t$ is the gradient or a moving average of the gradients (momentum).

one can use either the usual or the NTK parameterization, which consists of a layer-wise scaling of the parameters. The relevance of this property is discussed in Appendix E.1.

Compared to the standard regularization $\bar{\mathbf{H}} + \lambda \mathbf{I}$ and Nesterov's cubic regularization, the anisotropic Nesterov regularization does not break the property of invariance by subset-wise scaling of the parameters of 8. This is mainly due to our choice to keep only the diagonal coefficients of $\mathbf{D}_{\boldsymbol{\theta}}^{(3)}(\mathbf{u})$ while discarding the others. In particular, the off-diagonal coefficients contain cross-derivatives that would be difficult to include in an invariant training step.

## 5 EXPERIMENTS

### 5.1 EMPIRICAL COMPUTATION OF $\bar{\mathbf{H}}$ AND $\boldsymbol{\eta}$

As recalled in Section 2, many works perform a diagonal, block-diagonal or block-tridiagional (Martens & Grosse, 2015) approximation of the Hessian or its inverse. Since a summary $\bar{\mathbf{H}}$ of the Hessian and its inverse $\bar{\mathbf{H}}^{-1}$ are available and all their off-diagonal coefficients have been computed and kept, one can to check if these coefficients are indeed negligible.

**Setup.** We have trained LeNet-5 and VGG-11'[5] on CIFAR-10 using SGD with momentum. Before each epoch, we compute the full-batch gradient, denoted by $\mathbf{u}$, which we use as a direction to compute $\bar{\mathbf{H}}$, again in full-batch. We report submatrices of $\bar{\mathbf{H}}$ and $\bar{\mathbf{H}}^{-1}$ at initialization and at the epoch where the validation loss is the best in Figure 1a (LeNet) and Figure 1b (VGG-11').

For the sake of readability, $\bar{\mathbf{H}}$ has been divided into blocks: a weight-weight block $\bar{\mathbf{H}}_{\mathrm{WW}}$, a bias-bias block $\bar{\mathbf{H}}_{\mathrm{BB}}$, and a weight-bias block $\bar{\mathbf{H}}_{\mathrm{WB}}$. They represent the interactions between the layers: for instance, $(\bar{\mathbf{H}}_{\mathrm{WB}})_{l_1 l_2}$ represents the interaction between the tensor of weights of layer $l_1$ and the vector of biases of layer $l_2$.

**Results on $\bar{\mathbf{H}}$.** First, the block-diagonal approximation of the Hessian is indeed very rough, while the block-diagonal approximation of the inverse Hessian seems to be more reasonable (at least in these setups), which has already been shown by Martens & Grosse (2015). Second, there seem to be long-range interactions between layers, both at initialization and after several epochs. For LeNet, all the layers (except the first one) seem to interact together at initialization (Fig. 1a). In the matrix $\bar{\mathbf{H}}^{-1}$ computed on VGG, the last 3 layers interact strongly and the last 6 layers also interact, but a bit less.

According to these observations, a neural network should also be considered as a whole, in which layers can hardly be studied independently from each other. To our knowledge, this result is the first scalable representation of interactions between distant layers, based on second-order information.

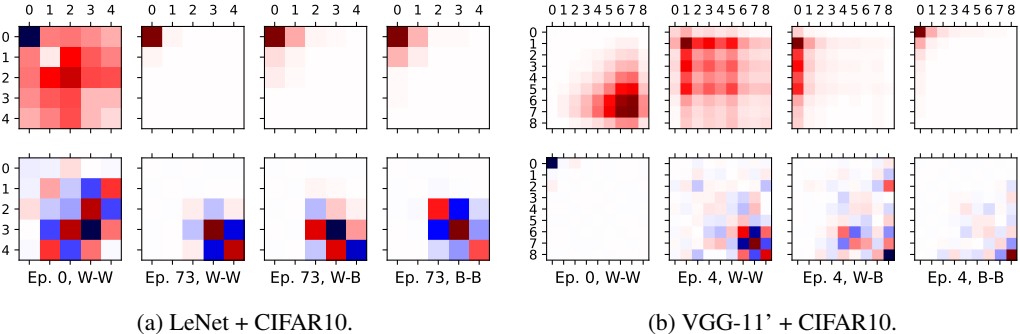

(a) LeNet + CIFAR10.  (b) VGG-11' + CIFAR10.

Figure 1: Setup: models trained by SGD on CIFAR-10. Submatrices of $\bar{\mathbf{H}}$ (first row) and $\bar{\mathbf{H}}^{-1}$ (second row), where focus is on interactions: weight-weight, weight-bias, bias-bias of the different layers, at initialization and before best validation loss epoch.

---

[5]VGG-11' is a variant of VGG-11 with only one fully-connected layer at the end, instead of 3.

**Results on $\eta_*$.** The evolution of the learning rates $\eta_*$ computed according to 10 in LeNet and VGG is shown in Figure 2b. First, the learning rates computed for the biases are larger than those computed for the weights. Second, even if only the weights are considered, the computed $\eta_*$ can differ by several orders of magnitude. Finally, the first two layers of LeNet (which are convolutional) have smaller $\eta_*$ than the last three layers (which are fully-connected). Conversely, in VGG, the weights of the last (convolutional) layers have a smaller $\eta_*$ than those of the first layers.

## 5.2 TRAINING EXPERIMENTS

In this section, we show a proof-of-concept of the optimization method 1 (summarized in Algorithm 1), on simple vision tasks and with medium-sized neural networks. All the implementation details are available in Appendix G. In particular, we have introduced a step size $\lambda_1$ that leads to the following modification of the training step 8: $\boldsymbol{\theta}_{t+1} = \boldsymbol{\theta}_t - \lambda_1 \mathbf{U}_t \mathbf{I}_{P:S} \boldsymbol{\eta}_t^*$.

---

**Algorithm 1** Informal implementation of the second-order method described in Section 4.

Let $u_t(\cdot)$ be a function computing a direction of descent $\mathbf{u}_t$ from a gradient $\mathbf{g}_t$ and $\mathbf{U}_t = \mathrm{Diag}(\mathbf{u}_t)$.

Hyperparameters: $\lambda, \lambda_{\mathrm{int}}$
$\mathcal{D}_{\mathrm{g}}, \mathcal{D}_{\mathrm{newt}}$ : independent samplers of minibatches
**for** $t \in [1, T]$ **do**
$\quad Z_t \sim \mathcal{D}_{\mathrm{g}}, \tilde{Z}_t \sim \mathcal{D}_{\mathrm{newt}}$ $\hfill$ (sample minibatches)
$\quad \mathbf{g}_t \leftarrow \frac{\mathrm{d}\mathcal{L}}{\mathrm{d}\boldsymbol{\theta}}(\boldsymbol{\theta}_t, Z_t)$ $\hfill$ (backward pass)
$\quad \mathbf{u}_t \leftarrow u_t(\mathbf{g}_t)$ $\hfill$ (custom direction of descent)
$\quad \bar{\mathbf{g}}_t \leftarrow \mathbf{D}_{\boldsymbol{\theta}_t}^{(1)}(\mathbf{u}_t) = \mathbf{I}_{S:P} \mathbf{U}_t \frac{\mathrm{d}\mathcal{L}}{\mathrm{d}\boldsymbol{\theta}}(\boldsymbol{\theta}_t, \tilde{Z}_t)$
$\quad \bar{\mathbf{H}}_t \leftarrow \mathbf{D}_{\boldsymbol{\theta}_t}^{(2)}(\mathbf{u}_t) = \mathbf{I}_{S:P} \mathbf{U}_t \frac{\mathrm{d}^2\mathcal{L}}{\mathrm{d}\boldsymbol{\theta}^2}(\boldsymbol{\theta}_t, \tilde{Z}_t) \mathbf{U}_t \mathbf{I}_{P:S}$
$\quad \mathbf{D}_t \leftarrow \mathrm{Diag}(|\mathbf{D}_{\boldsymbol{\theta}_t}^{(3)}(\mathbf{u}_t)|_{iii}^{1/3} : i \in \{1, \cdots, S\}) \in \mathbb{R}^{S^2}$
$\quad \boldsymbol{\eta}_t \leftarrow$ solution of $\boldsymbol{\eta} = \left(\bar{\mathbf{H}}_t + \frac{\lambda_{\mathrm{int}}}{2}\|\mathbf{D}_t\boldsymbol{\eta}\|\mathbf{D}_t^2\right)^{-1} \bar{\mathbf{g}}_t$ with max. norm $\|\mathbf{D}_t\boldsymbol{\eta}\|$ $\hfill$ (Method 1)
$\quad \boldsymbol{\theta}_{t+1} \leftarrow \boldsymbol{\theta}_t - \lambda \mathbf{U}_t \mathbf{I}_{P:S} \boldsymbol{\eta}_t$ $\hfill$ (training step)
**end for**

---

**Setup.** We consider 4 image classification setups:

- **MLP**: multilayer perceptron trained on MNIST with layers of sizes 1024, 200, 100, 10, and $\tanh$ activation;

- **LeNet**: LeNet-5 (LeCun et al., 1998) model trained on CIFAR-10 with 2 convolutional layers of sizes 6, 16, and 3 fully connected layers of sizes 120, 84, 10;

- **VGG**: VGG-11' trained on CIFAR-10. VGG-11' is a variant of VGG-11 (Simonyan & Zisserman, 2014) with only one fully-connected layer at the end, instead of 3, with ELU activation function (Clevert et al., 2015), without batch-norm;

- **BigMLP**: multilayer perceptron trained on CIFAR-10, with 20 layers of size 1024 and one classification layer of size 10, with ELU activation function.

And we have tested 3 optimization methods:

- **Adam**: the best learning rate has been selected by grid-search;

- **K-FAC**: the best learning rate and damping have been selected by grid-search;

- **NewtonSummary** (ours): the best $\lambda_1$ and $\lambda_{\mathrm{int}}$ have been selected by grid search.

**Results.** The evolution of the training loss is plotted in Figure 2a for each of the 3 optimization methods, for 5 different seeds. In each set of experiments, the training is successful, but slow or unstable at some points. Anyway, the minimum training loss achieved by Method 1 (NewtonSummary) is comparable to the minimum training loss achieved by K-KAC or Adam in all the series except for BigMLP, whose training is extremely slow. We provide the results on the test set in Appendix I and a comparison of the training times in Appendix M.

Some runs have encountered instabilities due to very large step sizes $\boldsymbol{\eta}_*$. In fact, we did not use any safeguards, such as a regularization term $\lambda\mathbf{I}$ added to $\bar{\mathbf{H}}$, or clipping the learning rates to avoid increasing the number of hyperparameters.

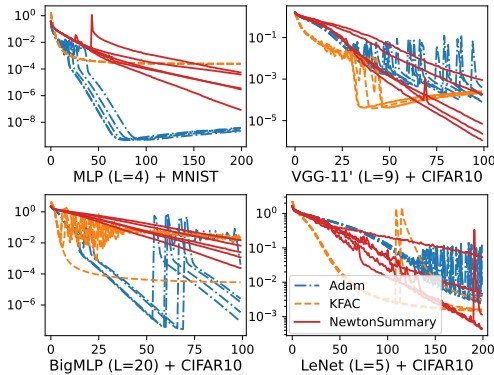

(a) Training curves in different setups. The reported loss is the negative log-likelihood computed on the training set.

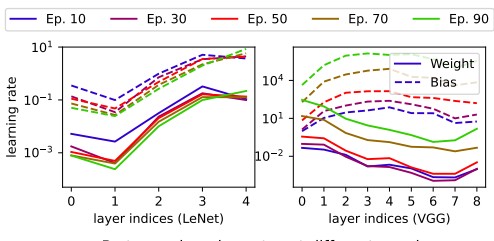

(b) Setup: LeNet, VGG-11' trained by SGD on CIFAR-10. Learning rates $\eta_*$ computed according to 10, specific to each tensor of weights and tensor of biases of each layer. For each epoch $k \in \{10, 30, 50, 70, 90\}$, the reported value has been averaged over the epochs $[k-10, k+9]$ to remove the noise.

Figure 2: Training curves: Method 1 (solid lines) versus its diagonal approximation (dotted lines) with various hyperparameters.

**Extension to very large models.** Since the matrix $\bar{\mathbf{H}}$ can be computed numerically as long as $S$ remains relatively small, this method may become unpractical for very large models. However, Method 1 is flexible enough to be adapted to such models: one can regroup tensors "of the same kind" to build a coarser partition of the parameters, and thus obtain a small $S$, which is exactly what is needed to compute $\bar{\mathbf{H}}$ and invert it. The difficulty would then be to find a good partition of the parameters, by grouping all the tensors that "look alike". We provide an example in Appendix H with a very deep multilayer perceptron.

**Choice of the partition.** We propose in Appendix J an empirical study and a discussion about the choice of the partition of the parameters. We show how it affects the training time and the final loss.

**Importance of the interactions between layers.** We show in Appendix K that the interactions between layers cannot be neglected when using our method: Method 1 outperforms its *diagonal approximation* on LeNet and VGG11', showing the importance of off-diagonal coefficients of $\bar{\mathbf{H}}$.

## 6 DISCUSSION

**Convergence rate.** Method 1 does not come with a precise convergence rate. The rate proposed in Appendix F (Theorem 1) gives only a heuristic. Given the convergence rates of Newton's method and Cauchy's steepest descent, we can expect to find some in-between convergence rates. Since Cauchy's steepest method is vulnerable to a highly anisotropic Hessian, it would be valuable to know how much this weakness is overcome by our method.

**Practicality.** Despite the interesting properties of Method 1 (scalability, invariance by reparameterization, evaluation of long-range interactions between layers), we have proposed nothing more than a proof-of-concept. This method remains subject to instabilities during training, which is expected for a second-order method, but not acceptable for the end user. Therefore, some additional tricks should be added to improve the stability of the training, which is a common practice, but usually comes with additional hyperparameters to tune. Also, improvements are needed to reduce the duration of each epoch, which is longer than with K-FAC.

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

# A  EXTENSIONS OF PEARLMUTTER'S TRICK

In this appendix, we show how to use the trick of Pearlmutter (1994) to compute the terms of the Taylor expansion of $\mathcal{L}$ and the tensor $\mathbf{D}_{\boldsymbol{\theta}}^d(\mathbf{u})$ defined in Eqn. 6.

**Fast computation of the terms of the Taylor expansion.**  We recall the Taylor expansion:

$$\mathcal{L}(\boldsymbol{\theta} + \mathbf{u}) = \mathcal{L}(\boldsymbol{\theta}) + \sum_{d=1}^{D} \frac{1}{d!} \frac{\mathrm{d}^d \mathcal{L}}{\mathrm{d}\boldsymbol{\theta}^d}(\boldsymbol{\theta})[\mathbf{u}, \cdots, \mathbf{u}] + o(\|\mathbf{u}\|^D). \tag{11}$$

We want to compute:

$$\tilde{\mathbf{D}}_{\boldsymbol{\theta}}^d(\mathbf{u}) := \frac{\mathrm{d}^d \mathcal{L}}{\mathrm{d}\boldsymbol{\theta}^d}(\boldsymbol{\theta})[\mathbf{u}, \cdots, \mathbf{u}] \in \mathbb{R}. \tag{12}$$

To do this, we will use:

$$\tilde{\mathbf{D}}_{\boldsymbol{\theta}}^{d-1}(\mathbf{u}) = \frac{\mathrm{d}^{d-1} \mathcal{L}}{\mathrm{d}\boldsymbol{\theta}^{d-1}}(\boldsymbol{\theta})[\mathbf{u}, \cdots, \mathbf{u}] \in \mathbb{R}. \tag{13}$$

We use the following recursion formula:

$$\tilde{\mathbf{D}}_{\boldsymbol{\theta}}^d(\mathbf{u}) = \left( \frac{\mathrm{d}\tilde{\mathbf{D}}_{\boldsymbol{\theta}}^{d-1}(\mathbf{u})}{\mathrm{d}\boldsymbol{\theta}} \right)^T \mathbf{u}. \tag{14}$$

Therefore, at each step $d$, we only have to compute the gradient of a scalar $\tilde{\mathbf{D}}_{\boldsymbol{\theta}}^d(\mathbf{u})$ according to $\boldsymbol{\theta} \in \mathbb{R}^P$, and compute a dot product in the space $\mathbb{R}^P$. So, computing $\tilde{\mathbf{D}}_{\boldsymbol{\theta}}^d(\mathbf{u})$ has a complexity proportional to $d \times P$, and does not require the computation of the full tensor $\frac{\mathrm{d}^d \mathcal{L}}{\mathrm{d}\boldsymbol{\theta}^d}(\boldsymbol{\theta}) \in \mathbb{R}^{P^d}$.

**Fast computation of $\mathbf{D}_{\boldsymbol{\theta}}^d(\mathbf{u})$.**  We assume that the parameter $\boldsymbol{\theta}$ is represented by a sequence of vectors $(\mathbf{T}^1, \cdots, \mathbf{T}^S)$: each coordinate $\theta_i$ belongs to exactly one of the $\mathbf{T}^s$. In the same way, given a direction $\mathbf{u} \in \mathbb{R}^P$ in the space of the parameters, $\mathbf{u}$ can be represented by a sequence of vectors $(\mathbf{U}^1, \cdots, \mathbf{U}^S)$.

We want to compute the tensor $\mathbf{D}_{\boldsymbol{\theta}}^d(\mathbf{u}) \in \mathbb{R}^{S^d}$, whose coefficients are:

$$(\mathbf{D}_{\boldsymbol{\theta}}^d(\mathbf{u}))_{s_1, \cdots, s_d} = \frac{\partial^d \mathcal{L}}{\partial \mathbf{T}^{s_1} \cdots \partial \mathbf{T}^{s_d}}(\boldsymbol{\theta})[\mathbf{U}^{s_1}, \cdots, \mathbf{U}^{s_d}], \tag{15}$$

for each multi-index $(s_1, \cdots, s_d) \in \{1, \cdots, S\}^d$.

Let us assume that $\mathbf{D}_{\boldsymbol{\theta}}^{d-1}(\mathbf{u})$ is available. We can compute the coefficients of $\mathbf{D}_{\boldsymbol{\theta}}^d(\mathbf{u})$ as follows:

$$(\mathbf{D}_{\boldsymbol{\theta}}^d(\mathbf{u}))_{s_1, \cdots, s_d} = \left( \frac{\partial (\mathbf{D}_{\boldsymbol{\theta}}^{d-1}(\mathbf{u}))_{s_1, \cdots, s_{d-1}}}{\partial \mathbf{T}^{s_d}} \right)^T \mathbf{U}^{s_d} \tag{16}$$

That way, the tensor $\mathbf{D}_{\boldsymbol{\theta}}^d(\mathbf{u})$ can be computed without having to compute the full derivative $\frac{\mathrm{d}^d \mathcal{L}}{\mathrm{d}\boldsymbol{\theta}^d}$. In fact, we do not need to store objects of size greater than $S^{d-1} \times P$: the last operation requires storing $\mathbf{D}_{\boldsymbol{\theta}}^{d-1}(\mathbf{u})$, which is of size $S^{d-1}$, and the gradient of each of its elements with respect to $(\mathbf{T}^1, \cdots, \mathbf{T}^S)$, which is of size $P$.

**Python implementation.**  We provide here an example of implementation of this extension of Pearlmutter's trick. First, we define in Alg. 2 the function `dercon`, standing for "derivation+contraction". Second, we provide in Alg. 3 the function `diff`, which outputs a list of tensors $(\mathbf{D}_{\boldsymbol{\theta}}^d(\mathbf{u}))_{d \in \{0, \cdots D\}}$.

We recall that $\mathbf{D}_{\boldsymbol{\theta}}^d(\mathbf{u}) \in \mathbb{R}^{S^d}$, where $s$ is the number of groups of tensors in $\boldsymbol{\theta}$, $\mathbf{u} \in \mathbb{R}^P$ is a direction in the parameter space $\mathbb{R}^P$, that can be represented as a tuple of groups of tensors, with the exact same structure as $\boldsymbol{\theta}$.

Also, to keep this implementation efficient for $d \geq 2$, we reduce the computational and storage cost of the tensors $\mathbf{D}_{\boldsymbol{\theta}}^d(\mathbf{u})$ by using the symmetry structure of $\mathbf{D}_{\boldsymbol{\theta}}^d(\mathbf{u})$. More specifically, for any permutation $\sigma : \{1, \cdots, d\} \to \{1, \cdots, d\}$:

$$\forall (i_1, \cdots, i_d), \quad (\mathbf{D}_{\boldsymbol{\theta}}^d(\mathbf{u}))_{i_1, \cdots, i_d} = (\mathbf{D}_{\boldsymbol{\theta}}^d(\mathbf{u}))_{i_{\sigma(1)}, \cdots, i_{\sigma(d)}}. \tag{17}$$

So, instead of computing and storing the full tensor $\mathbf{D}_{\boldsymbol{\theta}}^d(\mathbf{u}) \in\in \mathbb{R}^{S^d}$, we just have to store the coefficients $(\mathbf{D}_{\boldsymbol{\theta}}^d(\mathbf{u}))_{i_1, \cdots, i_d}$, with $i_1 \leq \cdots \leq i_d$. So, we compute and store only $\frac{(S+d-1)!}{d!(S-1)!}$ coefficients instead of $S^d$. For example, with $S = 10$ groups of parameters and $d = 3$, we store only 220 coefficients instead of $S^d = 1000$. We typically use this trick in Alg. 3.

---

**Algorithm 2** Implementation of the `dercon` function, which performs a derivation+contraction operation. This operation allows us to gather higher-order information without having to store large tensors.

---

```python
import torch
from . import select_params, dot

"""
Existing functions:
 * select_params(x, s0):
   returns the groups of params of x with indices s>=s0
 * dot(x1, x2):
   x1 and x2 are tuples of tuples of tensors with the same structure
     a) compute the dot-product between each pair of tensors (t1, t2)
     b) sum these inside ench group of tensors
     c) return the result (which is a torch.tensor containing as
     many coeffs as there are groups of params in x1 and x2
"""
def dercon(f, theta, u, s0):
    # Derivation + contraction

    # Derivation
    theta_s0 = select_params(theta, s0)
    deriv = torch.autograd.grad(f, theta_s0, create_graph = True)

    # Contraction
    u_s0 = select_params(u, s0)
    result = dot(deriv, u_s0)
    return result
```

---

**Algorithm 3** Implementation of the `diff` function, computing $\mathbf{D}^d$ iteratively at several orders. We denote by `L` the loss function, and by `(x, y)` a data point.

```python
import torch
from itertools import combinations_with_replacement
from . import dercon

def diff(L, x, y, order):
    lstD = [None]*(order+1)     # initialize the object to output

    # Forward pass
    loss = L(x, y)
    # Store the loss (without keeping the computational graph)
    lstD[0] = loss.detach()

    # Compute the derivative of the loss and contract it
    # in the directions given by the groups of params of u.
    deriv = {tuple(): dercon(loss, u, 0)}
    # Store the result (without keeping the computational graph)
    lstD[1] = {k: v.detach() for k, v in deriv.items()}

    # Computations with d >= 2
    for d in range(2, order+1):
        new_deriv = {}
        # We only compute the derivatives of certain coefficients
        # of the matrix D^{d-1}: thanks to symmetries of D^{d-1},
        # it is not necessary to compute all of them.

        # Create the list of indices of the relevant coefficients
        lst_idx = [tuple(sorted(idx)) for idx in \
            combinations_with_replacement(range(S), d-1)]

        # Compute these coefficients
        for idx in lst_idx:
            # we derive the coeff of D^{d-1} with multi-index idx

            init, last = idx[:-1], idx[-1]
            imax = last if len(init)==0 else last-init[-1]

            # To access that coeff, we use deriv[init][imax]
            new_deriv[idx] = dercon(deriv[init][imax], u, last)
        # Store the result (without keeping the comp. graph)
        lstD[d] = {k: v.detach() for k, v in new_deriv.items()}
        # Prepare the next iteration
        deriv = new_deriv
    return lstD
```

## B   DERIVATION OF THE SECOND-ORDER METHOD

We consider an update of $\boldsymbol{\theta}$ with one learning rate $\eta_s$ for each subset $\mathcal{I}_s$ of parameters. Let $\mathbf{I}_{S:P} \in \mathbb{R}^{S \times P}$ be the *partition matrix*, verifying $(\mathbf{I}_{S:P})_{sp} = 1$ if $p \in \mathcal{I}_s$ and 0 otherwise, and $\mathbf{I}_{P:S} := \mathbf{I}_{S:P}^T$. We consider an update based on a given direction $\mathbf{u}_t$ and we define $\mathbf{U}_t := \mathrm{Diag}(\mathbf{u}_t)$:

$$\boldsymbol{\theta}_{t+1} = \boldsymbol{\theta}_t - \mathbf{U}_t \mathbf{I}_{P:S} \boldsymbol{\eta}, \tag{18}$$

where $\boldsymbol{\eta} = (\eta_1, \cdots, \eta_S) \in \mathbb{R}^S$.

The second-order approximation of $\mathcal{L}$ gives:

$$\begin{aligned}
\mathcal{L}(\boldsymbol{\theta}_{t+1}) &= \mathcal{L}(\boldsymbol{\theta}_t - \mathbf{U}_t \mathbf{I}_{P:S} \boldsymbol{\eta}) \\
&= \mathcal{L}(\boldsymbol{\theta}_t) - \boldsymbol{\eta}^T \mathbf{I}_{S:P} \mathbf{U}_t \frac{\mathrm{d}\mathcal{L}}{\mathrm{d}\boldsymbol{\theta}}(\boldsymbol{\theta}_t) \\
&\quad + \frac{1}{2} \boldsymbol{\eta}^T \mathbf{I}_{S:P} \mathbf{U}_t \frac{\mathrm{d}^2\mathcal{L}}{\mathrm{d}\boldsymbol{\theta}^2}(\boldsymbol{\theta}_t) \mathbf{U}_t \mathbf{I}_{P:S} \boldsymbol{\eta} + o(\|\boldsymbol{\eta}\|^2) \\
&= \mathcal{L}(\boldsymbol{\theta}_t) - \boldsymbol{\eta}^T \mathbf{I}_{S:P} \mathbf{U}_t \mathbf{g}_t \\
&\quad + \frac{1}{2} \boldsymbol{\eta}^T \mathbf{I}_{S:P} \mathbf{U}_t \mathbf{H}_t \mathbf{U}_t \mathbf{I}_{P:S} \boldsymbol{\eta} + o(\|\boldsymbol{\eta}\|^2) \\
&= \mathcal{L}(\boldsymbol{\theta}_t) - \boldsymbol{\eta}^T \bar{\mathbf{g}}_t + \frac{1}{2} \boldsymbol{\eta}^T \bar{\mathbf{H}}_t \boldsymbol{\eta} + o(\|\boldsymbol{\eta}\|^2),
\end{aligned}$$

where:

$$\bar{\mathbf{g}}_t := \mathbf{I}_{S:P} \mathbf{U}_t \mathbf{g}_t \in \mathbb{R}^S, \qquad \bar{\mathbf{H}}_t := \mathbf{I}_{S:P} \mathbf{U}_t \mathbf{H}_t \mathbf{U}_t \mathbf{I}_{P:S} \in \mathbb{R}^{S \times S}. \tag{19}$$

Now, we omit the $o(\|\boldsymbol{\eta}\|^2)$ term and we want to minimize according to $\boldsymbol{\eta}$ the variation of the loss:

$$\mathcal{L}(\boldsymbol{\theta}_{t+1}) - \mathcal{L}(\boldsymbol{\theta}_t) \approx \boldsymbol{\Delta}_2(\boldsymbol{\eta}) := \mathcal{L}(\boldsymbol{\theta}_t) - \boldsymbol{\eta}^T \bar{\mathbf{g}}_t + \frac{1}{2} \boldsymbol{\eta}^T \bar{\mathbf{H}}_t \boldsymbol{\eta}. \tag{20}$$

We have: $\frac{\mathrm{d}\boldsymbol{\Delta}_2}{\mathrm{d}\boldsymbol{\eta}} = -\bar{\mathbf{g}}_t + \bar{\mathbf{H}}_t \boldsymbol{\eta}$, which is zero if, and only if: $\bar{\mathbf{g}}_t = \bar{\mathbf{H}}_t \boldsymbol{\eta}$. If this linear system can be inverted, one can choose:

$$\boldsymbol{\eta} = \boldsymbol{\eta}_t^* := \bar{\mathbf{H}}_t^{-1} \bar{\mathbf{g}}_t. \tag{21}$$

**Interpretation as multivariate optimization.**   This method can also be derived by taking the point of view of multivariate optimization. Within our setup, $\boldsymbol{\theta}$ is considered as a tuple of tensors $(\mathbf{T}^1, \cdots, \mathbf{T}^S)$. Thus, we want to minimize the loss $\mathcal{L}$ with respect to $(\mathbf{T}^1, \cdots, \mathbf{T}^S)$. By abuse of notation, we will write:

$$\mathcal{L}(\boldsymbol{\theta}) = \mathcal{L}(\mathbf{T}^1, \cdots, \mathbf{T}^S). \tag{22}$$

Now, we assume that we dispose of a direction of descent $-\mathbf{U}^s$ for each $\mathbf{T}^s$. Thus, we can consider the following training step:

$$\forall s \in \{1, \cdots S\}, \quad \mathbf{T}^s \leftarrow \mathbf{T}^s - \eta_s \mathbf{U}^s, \tag{23}$$

where $(\eta_1, \cdots \eta_S)$ are learning rates. Thus, the loss change after the training step is:

$$f(\eta_1, \cdots, \eta_S) := \mathcal{L}(\mathbf{T}^1 - \eta_1 \mathbf{U}^1, \cdots, \mathbf{T}^S - \eta_S \mathbf{U}^S) - \mathcal{L}(\mathbf{T}^1, \cdots, \mathbf{T}^S). \tag{24}$$

When we do a second-order approximation, the loss change becomes:

$$f(\eta_1, \cdots, \eta_S) \approx -\sum_{s=1}^{S} \eta_s \left(\frac{\partial\mathcal{L}}{\partial\mathbf{T}^s}\right)^T \mathbf{U}^s + \frac{1}{2} \sum_{s_1=1}^{S} \sum_{s_2=1}^{S} \eta_{s_1} \eta_{s_2} (\mathbf{U}^{s_1})^T \frac{\partial^2\mathcal{L}}{\partial\mathbf{T}^{s_1}\partial\mathbf{T}^{s_2}} \mathbf{U}^{s_2} \tag{25}$$

$$\approx -\boldsymbol{\eta}^T \bar{\mathbf{g}} + \frac{1}{2} \boldsymbol{\eta}^T \bar{\mathbf{H}} \boldsymbol{\eta}, \tag{26}$$

where $\boldsymbol{\eta} = (\eta_1, \cdots, \eta_S) \in \mathbb{R}^S$, $\bar{\mathbf{g}} \in \mathbb{R}^S$ is the gradient of $f$ and $\bar{\mathbf{H}} \in \mathbb{R}^{S \times S}$ is the Hessian of $f$:

$$\bar{g}_s = \frac{\partial f}{\partial \eta_s} = \left(\frac{\partial\mathcal{L}}{\partial\mathbf{T}^s}\right)^T \mathbf{U}^s \qquad \bar{H}_{s_1 s_2} = \frac{\partial^2 f}{\partial\eta_{s_1}\partial\eta_{s_2}} = (\mathbf{U}^{s_1})^T \frac{\partial^2\mathcal{L}}{\partial\mathbf{T}^{s_1}\partial\mathbf{T}^{s_2}} \mathbf{U}^{s_2}. \tag{27}$$

Finally, one can minimize the order-2 approximation of $f$ (Eqn. 26) with respect to $\boldsymbol{\eta}$, with any numerical or analytical technique. If done analytically, we roll back to Eqn. 21.

**Interpretation as optimization constrained to a vector subspace.** In addition, the training step can be interpreted as an optimization of the descent direction within a vector subspace. We assume that we dispose of a direction of descent $-\mathbf{u} \in \mathbb{R}^P$. Let $(\mathbf{u}_1, \cdots, \mathbf{u}_S)$ be a family of vectors of $\mathbb{R}^P$ defined by:

$$\forall s, \quad \mathbf{u}_s = (0_{P_1}, \cdots, 0_{P_{s-1}}, \mathbf{U}^s, 0_{P_{s+1}}, \cdots, 0_{P_S}), \tag{28}$$

where $P_i$ is the size of the tensor $\mathbf{U}^i$ (or of $\mathbf{T}^i$) and $0_{P_i}$ is the null tensor of size $P_i$.

We want to minimize the loss decrease after one training step with respect to the step of descent $\mathbf{v}$, which is assumed to be small:

$$g(\mathbf{v}) = \mathcal{L}(\boldsymbol{\theta} - \mathbf{v}) - \mathcal{L}(\boldsymbol{\theta}), \tag{29}$$

under the condition $\mathbf{v} \in \mathrm{span}(\mathbf{u}_1, \cdots, \mathbf{u}_S)$. An order-2 approximation of $g$ gives:

$$g(\mathbf{v}) \approx \tilde{g}(\mathbf{v}) := -\mathbf{v}^T \frac{\mathrm{d}\mathcal{L}}{\mathrm{d}\boldsymbol{\theta}}(\boldsymbol{\theta}) + \frac{1}{2}\mathbf{v}^T \frac{\mathrm{d}^2\mathcal{L}}{\mathrm{d}\boldsymbol{\theta}^2}(\boldsymbol{\theta})\mathbf{v}.$$

Now, we look for the vector $\mathbf{v}^*$ such that:

$$\mathbf{v}^* = \underset{\mathbf{v}^* \in \mathrm{span}(\mathbf{u}_1, \cdots, \mathbf{u}_S)}{\arg\min} \left( -\mathbf{v}^T \frac{\mathrm{d}\mathcal{L}}{\mathrm{d}\boldsymbol{\theta}}(\boldsymbol{\theta}) + \frac{1}{2}\mathbf{v}^T \frac{\mathrm{d}^2\mathcal{L}}{\mathrm{d}\boldsymbol{\theta}^2}(\boldsymbol{\theta})\mathbf{v} \right). \tag{30}$$

The solution is given by:

$$\mathbf{v}^* := \mathbf{U}\mathbf{I}_{P:S}\bar{\mathbf{H}}^{-1}\bar{\mathbf{g}} = \mathbf{U}\mathbf{I}_{P:S}\boldsymbol{\eta}^*, \tag{31}$$

where $\mathbf{U} = \mathrm{Diag}(\mathbf{u})$. So, we recover the method described in Section 4.1.

## C LINK WITH CAUCHY'S STEEPEST DESCENT AND NEWTON'S METHOD

**Cauchy's steepest descent.** Let us consider the trivial partition: $S = 1$, $\mathcal{I}_1 = \{1, \cdots, P\}$. So, $\mathbf{I}_{S:P} = (1, \cdots, 1) = \mathbb{1}_S^T$. Therefore, the training step is:

$$\boldsymbol{\theta}_{t+1} := \boldsymbol{\theta}_t - \mathbf{G}_t\mathbb{1}_S(\mathbb{1}_S^T\mathbf{G}_t\mathbf{H}_t\mathbf{G}_t\mathbb{1}_S)^{-1}\mathbb{1}_S^T\mathbf{G}_t\mathbf{g}_t = \boldsymbol{\theta}_t - \mathbf{g}_t\frac{\mathbf{g}_t^T\mathbf{g}_t}{\mathbf{g}_t^T\mathbf{H}_t\mathbf{g}_t}, \tag{32}$$

since $\mathbf{G}_t\mathbb{1}_S = \mathbf{g}_t$. We recover Cauchy's steepest descent.

**Newton's method.** Since we aim to recover Newton's method, we assume that the Hessian $\mathbf{H}_t$ is positive definite. Let us consider the discrete partition: $S = P$, $\mathcal{I}_s = \{s\}$. So, $\mathbf{I}_{S:P} = \mathbf{I}_P$, the identity matrix of $\mathbb{R}^{P \times P}$. Therefore, the training step is:

$$\boldsymbol{\theta}_{t+1} := \boldsymbol{\theta}_t - \mathbf{G}_t(\mathbf{G}_t\mathbf{H}_t\mathbf{G}_t)^{-1}\mathbf{G}_t\mathbf{g}_t. \tag{33}$$

To perform the training step, we have to find $\mathbf{x} \in \mathbb{R}^P$ such that: $(\mathbf{G}_t\mathbf{H}_t\mathbf{G}_t)^{-1}\mathbf{G}_t\mathbf{g}_t = \mathbf{x}$. That is, solve the linear system $\mathbf{G}_t\mathbf{H}_t\mathbf{G}_t\mathbf{x} = \mathbf{G}_t\mathbf{g}_t$. In the case where all the coordinates of the gradient $\mathbf{g}_t$ are nonzero, we can write:

$$\mathbf{x} = \mathbf{G}_t^{-1}\mathbf{H}_t^{-1}\mathbf{G}_t^{-1}\mathbf{G}_t\mathbf{g}_t = \mathbf{G}_t^{-1}\mathbf{H}_t^{-1}\mathbf{g}_t, \tag{34}$$

so the training step becomes:

$$\boldsymbol{\theta}_{t+1} := \boldsymbol{\theta}_t - \mathbf{G}_t\mathbf{x} = \boldsymbol{\theta}_t - \mathbf{H}_t^{-1}\mathbf{g}_t, \tag{35}$$

which corresponds to Newton's method.

## D ANISOTROPIC NESTEROV CUBIC REGULARIZATION

Let $\mathbf{D}$ be a diagonal matrix whose diagonal coefficients are all strictly positive: $\mathbf{D} = \mathrm{Diag}(d_1, \cdots, d_S)$, with $d_i > 0$ for all $i$.

We want to minimize the function:

$$T(\boldsymbol{\eta}) := -\boldsymbol{\eta}^T\bar{\mathbf{g}} + \frac{1}{2}\boldsymbol{\eta}\bar{\mathbf{H}}\boldsymbol{\eta} + \frac{\lambda_{\mathrm{int}}}{6}\|\mathbf{D}\boldsymbol{\eta}\|^3. \tag{36}$$

The function $T$ is strictly convex if, and only if, $\bar{\mathbf{H}}$ is positive definite. Moreover, $T$ is differentiable twice and has at least one global minimum $\boldsymbol{\eta}_*$, so $\frac{\mathrm{d}T}{\mathrm{d}\boldsymbol{\eta}}(\boldsymbol{\eta}_*) = 0$. Therefore, we first look for the solutions of the equation $\frac{\mathrm{d}T}{\mathrm{d}\boldsymbol{\eta}}(\boldsymbol{\eta}) = 0$.

We have:

$$\frac{\mathrm{d}T}{\mathrm{d}\boldsymbol{\eta}}(\boldsymbol{\eta}) = -\bar{\mathbf{g}} + \bar{\mathbf{H}}\boldsymbol{\eta} + \frac{\lambda_{\text{int}}}{2}\|\mathbf{D}\boldsymbol{\eta}\|\mathbf{D}^2\boldsymbol{\eta}$$

$$= -\bar{\mathbf{g}} + \left(\bar{\mathbf{H}} + \frac{\lambda_{\text{int}}}{2}\|\mathbf{D}\boldsymbol{\eta}\|\mathbf{D}^2\right)\boldsymbol{\eta},$$

which is equal to zero if, and only if:

$$\bar{\mathbf{g}} = \left(\bar{\mathbf{H}} + \frac{\lambda_{\text{int}}}{2}\|\mathbf{D}\boldsymbol{\eta}\|\mathbf{D}^2\right)\boldsymbol{\eta}. \tag{37}$$

Let $\boldsymbol{\eta}' := \mathbf{D}\boldsymbol{\eta}$. Eqn. 37 is then equivalent to:

$$\bar{\mathbf{g}} = \left(\bar{\mathbf{H}}\mathbf{D}^{-1} + \frac{\lambda_{\text{int}}}{2}\|\boldsymbol{\eta}'\|\mathbf{D}\right)\boldsymbol{\eta}'.$$

$$= \frac{\lambda_{\text{int}}}{2}\mathbf{D}\left(\frac{2}{\lambda_{\text{int}}}\mathbf{D}^{-1}\bar{\mathbf{H}}\mathbf{D}^{-1} + \|\boldsymbol{\eta}'\|\mathbf{I}\right)\boldsymbol{\eta}'$$

Let $\mathbf{K} := \frac{2}{\lambda_{\text{int}}}\mathbf{D}^{-1}\bar{\mathbf{H}}\mathbf{D}^{-1}$. We want to solve:

$$\bar{\mathbf{g}} = \frac{\lambda_{\text{int}}}{2}\mathbf{D}\left(\mathbf{K} + \|\boldsymbol{\eta}'\|\mathbf{I}\right)\boldsymbol{\eta}' \tag{38}$$

Since $\mathbf{K}$ is positive definite if, and only if, $\bar{\mathbf{H}}$ is positive definite, we consider the following cases.

**Case 1: $\bar{\mathbf{H}}$ is positive definite.** In this case, Eqn. 38 is equivalent to:

$$\boldsymbol{\eta}' = \frac{2}{\lambda_{\text{int}}}\left(\mathbf{K} + \|\boldsymbol{\eta}'\|\mathbf{I}\right)^{-1}\mathbf{D}^{-1}\bar{\mathbf{g}}.$$

Now, let $r = \|\boldsymbol{\eta}'\|$. We want to solve:

$$r = \frac{2}{\lambda_{\text{int}}}\left\|\left(\mathbf{K} + r\mathbf{I}\right)^{-1}\mathbf{D}^{-1}\bar{\mathbf{g}}\right\|. \tag{39}$$

Trivially: $\boldsymbol{\eta}$ solution of 37 $\Rightarrow \mathbf{D}\boldsymbol{\eta}$ solution of 38 $\Rightarrow \|\mathbf{D}\boldsymbol{\eta}\|$ solution of 39. Reciprocally: $r$ solution of 39 $\Rightarrow \boldsymbol{\eta}' := (\bar{\mathbf{H}}\mathbf{D}^{-1} + \frac{\lambda_{\text{int}}}{2}r\mathbf{D})^{-1}\bar{\mathbf{g}}$ solution of 38 $\Rightarrow \mathbf{D}^{-1}\boldsymbol{\eta}'$ solution of 37.

Therefore, in order to find the unique global minimum of $T$, it is sufficient to solve Eqn. 39. This is doable numerically.

**Case 2: $\bar{\mathbf{H}}$ is not positive definite.** We follow the procedure proposed in (Nesterov & Polyak, 2006, Section 5). Let $\lambda_{\min}$ be the minimum eigenvalue of $\mathbf{K}$. So, $\lambda_{\min} \leq 0$. Following Nesterov & Polyak (2006), we look for the unique $\boldsymbol{\eta}'$ belonging to $\mathcal{C} := \{\boldsymbol{\eta}' \in \mathbb{R}^S : \|\boldsymbol{\eta}'\| > |\lambda_{\min}|\}$, which is also the solution of maximum norm of Eqn. 38. Conditionally to $\boldsymbol{\eta}' \in \mathcal{C}$, $(\mathbf{K} + \|\boldsymbol{\eta}'\|\mathbf{I})$ is invertible. So we only need to solve:

$$r > |\lambda_{\min}| : \quad r = \frac{2}{\lambda_{\text{int}}}\left\|\left(\mathbf{K} + r\mathbf{I}\right)^{-1}\mathbf{D}^{-1}\bar{\mathbf{g}}\right\|, \tag{40}$$

which has exactly one solution $r_*$. Then, we compute $\boldsymbol{\eta}_* := \mathbf{D}^{-1}(\bar{\mathbf{H}}\mathbf{D}^{-1} + \frac{\lambda_{\text{int}}}{2}r_*\mathbf{D})^{-1}\bar{\mathbf{g}}$.

# E  INVARIANCE BY SUBSET-WISE AFFINE REPARAMETERIZATION

## E.1  MOTIVATION

The choice of the best per-layer parameterization is still a debated question. On the theoretical side, the standard parameterization cannot be used to train very wide networks, because it leads to a

diverging first gradient step Yang & Hu (2021). Besides, the NTK parameterization is widely used in theoretical works in order to manage the infinite-width limit Jacot et al. (2018); Du et al. (2019); Arora et al. (2019); Lee et al. (2019); Mei & Montanari (2022). On the practical side, the standard parameterization is preferred over the NTK one because it leads to better results, both in terms of training and generalization.

So, there is no consensus about the best layer-wise parameterization. Thus, ensuring that a method is invariant by layer-wise affine reparameterization guarantees that its behavior remains the same whatever the choice of the user (standard or NTK parameterization).

### E.2 CLAIM

We consider a parameter $\tilde{\boldsymbol{\theta}}$ such that $\boldsymbol{\theta} = \varphi(\tilde{\boldsymbol{\theta}})$, where $\varphi$ is an invertible map, affine on each subset of parameters. Therefore, its Jacobian is: $\mathbf{J} = \mathrm{Diag}(\alpha_1, \cdots, \alpha_p)$, where, for all $1 \leq s \leq S$ and $1 \leq p_1, p_2 \leq P$, we have:

$$p_1, p_2 \in \mathcal{I}_s \Rightarrow \alpha_{p_1} = \alpha_{p_2} =: a_s. \tag{41}$$

Also, let $\bar{\mathbf{J}} = \mathrm{Diag}(a_1, \cdots, a_S)$.

We want to compare the training trajectory of $\mathcal{L}(\boldsymbol{\theta})$ and $\mathcal{L}(\varphi(\tilde{\boldsymbol{\theta}}))$ when using Method 1. For any quantity $\mathbf{x}$ computed with the parameterization $\boldsymbol{\theta}$, we denote by $\tilde{\mathbf{x}}$ its counterpart computed with the parameterization $\tilde{\boldsymbol{\theta}}$.

We compute $\tilde{\boldsymbol{\eta}}_*$. Equation 10 gives:

$$\tilde{\boldsymbol{\eta}}_* = \left( \tilde{\bar{\mathbf{H}}} + \frac{\lambda_{\mathrm{int}}}{2} \|\tilde{\mathbf{D}}\tilde{\boldsymbol{\eta}}_*\| \tilde{\mathbf{D}}^2 \right)^{-1} \tilde{\bar{\mathbf{g}}}. \tag{42}$$

Besides:

$$\tilde{\bar{\mathbf{H}}} := \mathbf{I}_{S:P} \tilde{\mathbf{U}} \tilde{\mathbf{H}} \tilde{\mathbf{U}} \mathbf{I}_{P:S}, \qquad \tilde{\bar{\mathbf{g}}} := \mathbf{I}_{S:P} \tilde{\mathbf{U}} \tilde{\mathbf{g}}.$$

To go further, we need to do an assumption about the direction $\mathbf{u}$.

**Assumption 1.** *We assume that $\mathbf{U}_t$ is computed in such a way that $\tilde{\mathbf{U}}_t = \mathbf{J} \mathbf{U}_t$ at every step.*

This assumption holds typically when $\mathbf{u}_t$ is the gradient at time step $t$. It holds also when $\mathbf{u}_t$ is a linear combination of past gradients:

$$\mathbf{u}_1 := \mathbf{g}_1, \qquad \mathbf{u}_{t+1} := \mu \mathbf{u}_t + \mu' \mathbf{g}_{t+1},$$

which includes the momentum.

To summarize, we have:

$$\tilde{\mathbf{U}} = \mathbf{J} \mathbf{U}, \quad \tilde{\mathbf{H}} = \mathbf{J} \mathbf{H} \mathbf{J}, \quad \tilde{\mathbf{g}} = \mathbf{J} \mathbf{g},$$

So:

$$\tilde{\bar{\mathbf{H}}} = \tilde{\mathbf{J}}^2 \mathbf{I}_{S:P} \mathbf{U} \mathbf{H} \mathbf{U} \mathbf{I}_{P:S} \tilde{\mathbf{J}}^2 = \tilde{\mathbf{J}}^2 \bar{\mathbf{H}} \tilde{\mathbf{J}}^2,$$
$$\tilde{\bar{\mathbf{g}}} = \tilde{\mathbf{J}}^2 \mathbf{I}_{S:P} \mathbf{U} \mathbf{g} = \tilde{\mathbf{J}}^2 \bar{\mathbf{g}},$$

since $\mathbf{J}$ and $\mathbf{U}$ are diagonal. And, since $\mathbf{D}_{ii} = |(\mathbf{D}_{\boldsymbol{\theta}}^{(3)}(\mathbf{u}))_{iii}|^{1/3}$, then $\tilde{\mathbf{D}}_{ii} = a_i^2 \mathbf{D}_{ii}$, thus $\tilde{\mathbf{D}} = \tilde{\mathbf{J}}^2 \mathbf{D}$.

Thus, Eqn. 42 becomes:

$$\tilde{\boldsymbol{\eta}}_* = \left( \tilde{\mathbf{J}}^2 \bar{\mathbf{H}} \tilde{\mathbf{J}}^2 + \frac{\lambda_{\mathrm{int}}}{2} \|\tilde{\mathbf{J}}^2 \mathbf{D} \tilde{\boldsymbol{\eta}}_*\| \tilde{\mathbf{J}}^4 \mathbf{D}^2 \right)^{-1} \tilde{\mathbf{J}}^2 \bar{\mathbf{g}},$$

which can be rewritten (since $\tilde{\mathbf{J}}$ is invertible):

$$\tilde{\mathbf{J}}^2 \tilde{\boldsymbol{\eta}}_* = \left( \bar{\mathbf{H}} + \frac{\lambda_{\mathrm{int}}}{2} \|\mathbf{D} \tilde{\mathbf{J}}^2 \tilde{\boldsymbol{\eta}}_*\| \mathbf{D}^2 \right)^{-1} \bar{\mathbf{g}}.$$

Therefore, $\tilde{\boldsymbol{\eta}}_*$ is a solution of Eqn. 10 in the parameterization $\tilde{\boldsymbol{\theta}}$ if, and only if, $\tilde{\mathbf{J}}^2\tilde{\boldsymbol{\eta}}_*$ is a solution in the parameterization $\boldsymbol{\theta}$. Moreover, $\|\tilde{\mathbf{D}}\tilde{\boldsymbol{\eta}}_*\| = \|\mathbf{D}\mathbf{J}^2\tilde{\boldsymbol{\eta}}_*\|$, so $\tilde{\boldsymbol{\eta}}_*$ is the solution of maximum norm $\|\tilde{\mathbf{D}}\tilde{\boldsymbol{\eta}}_*\|$ of 10 with parameterization $\tilde{\boldsymbol{\theta}}$ iff $\tilde{\mathbf{J}}^2\tilde{\boldsymbol{\eta}}_*$ is a solution of maximum norm $\|\mathbf{D}\mathbf{J}^2\tilde{\boldsymbol{\eta}}_*\|$ of 10 with parameterization $\boldsymbol{\theta}$.

Thus, $\boldsymbol{\eta}_* = \tilde{\mathbf{J}}^2\tilde{\boldsymbol{\eta}}_*$, and the update step in parameterization $\tilde{\boldsymbol{\theta}}$ is:
$$\tilde{\boldsymbol{\theta}}_{t+1} = \tilde{\boldsymbol{\theta}}_t - \tilde{\mathbf{U}}_t\mathbf{I}_{P:S}\tilde{\boldsymbol{\eta}}_*$$
$$= \tilde{\boldsymbol{\theta}}_t - \tilde{\mathbf{U}}_t\mathbf{I}_{P:S}\tilde{\mathbf{J}}^{-2}\boldsymbol{\eta}_*,$$
which can be rewritten:
$$\mathbf{J}^{-1}\boldsymbol{\theta}_{t+1} = \mathbf{J}^{-1}\boldsymbol{\theta}_t - \mathbf{U}\mathbf{J}\mathbf{I}_{P:S}\tilde{\mathbf{J}}^{-2}\boldsymbol{\eta}_*, \tag{43}$$
since $\varphi$ is an affine function with factor $\mathbf{J}$. Finally, Eqn. 43 boils down to:
$$\boldsymbol{\theta}_{t+1} = \boldsymbol{\theta}_t - \mathbf{U}\mathbf{I}_{P:S}\boldsymbol{\eta}_*, \tag{44}$$
which is exactly Method 1 in parameterization $\boldsymbol{\theta}$.

# F CONVERGENCE RATE IN A SIMPLE CASE

We study the convergence of the method presented in Section 4.1 (without anisotropic Nesterov's cubic regularization):
$$\boldsymbol{\theta}_{t+1} = \boldsymbol{\theta}_t - \mathbf{U}_t\mathbf{I}_{P:S}\boldsymbol{\eta}_t, \qquad\qquad \boldsymbol{\eta}_t := \bar{\mathbf{H}}_t^{-1}\bar{\mathbf{g}}_t, \tag{45}$$
where:
$$\bar{\mathbf{H}}_t := \mathbf{I}_{S:P}\mathbf{U}_t\mathbf{H}_t\mathbf{U}_t\mathbf{I}_{P:S}, \qquad\qquad \bar{\mathbf{g}}_t := \mathbf{I}_{S:P}\mathbf{U}_t\mathbf{g}_t,$$
$$\mathbf{H}_t := \frac{\mathrm{d}^2\mathcal{L}}{\mathrm{d}\boldsymbol{\theta}^2}(\boldsymbol{\theta}_t), \qquad\qquad \mathbf{g}_t := \frac{\mathrm{d}\mathcal{L}}{\mathrm{d}\boldsymbol{\theta}}(\boldsymbol{\theta}_t),$$
$$\mathbf{U}_t := -\mathbf{G}_t,$$
that is, the direction $\mathbf{u}_t$ is given by the gradient $\mathbf{g}_t$.

We study this optimization method in the case where $\mathcal{L}$ is a positive quadratic form:
$$\mathcal{L}(\boldsymbol{\theta}) := \frac{1}{2}\boldsymbol{\theta}^T\mathbf{H}\boldsymbol{\theta}, \tag{46}$$
where $\mathbf{H}$ is positive definite and block-diagonal: $\mathbf{H} = \mathrm{Diag}(\mathbf{H}_1, \cdots, \mathbf{H}_S)$.

We consider a partition $(\mathcal{I}_s)_{1 \le s \le S}$ of the parameter space consistent with the block-diagonal structure of $\mathbf{H}$. In other words, if the coefficient $H_{pp}$ of $\mathbf{H}$ lies in the submatrix $\mathbf{H}_s$, then $p \in \mathcal{I}_s$.

**Theorem 1.** *The method has a linear rate of convergence. For any $\boldsymbol{\theta}_t \ne 0$:*
$$\frac{\mathcal{L}(\boldsymbol{\theta}_{t+1})}{\mathcal{L}(\boldsymbol{\theta}_t)} \le \max_s\left(\frac{(A_s - a_s)^2}{(A_s + a_s)^2}\right),$$
*where $a_s = \min\mathrm{Sp}(\mathbf{H}_s)$ and $A_s = \max\mathrm{Sp}(\mathbf{H}_s)$. Moreover, this rate is optimal, since it is possible to build $\boldsymbol{\theta}_t$ such that:*
$$\frac{\mathcal{L}(\boldsymbol{\theta}_{t+1})}{\mathcal{L}(\boldsymbol{\theta}_t)} = \max_s\left(\frac{(A_s - a_s)^2}{(A_s + a_s)^2}\right).$$

Alternatively:
$$\frac{\mathcal{L}(\boldsymbol{\theta}_{t+1})}{\mathcal{L}(\boldsymbol{\theta}_t)} \le \max_s\left(\frac{(\gamma_s - 1)^2}{(\gamma_s + 1)^2}\right),$$
where $\gamma_s = A_s/a_s \ge 1$.

**Remark 1.** *For a given $\mathbf{H}$, better convergence rates can be achieved by reducing the $(\gamma_s)_s$, that is, choosing partitions $(\mathcal{I}_s)_s$ such that, for all $s$, the eigenvalues $(h_p)_{p \in \mathcal{I}_s}$ are not too spread out.*

In other words, good partitions are partitions such that indices of eigenvalues close to each other are grouped inside the same subset $\mathcal{I}_s$. On the contrary, grouping the parameters regardless of the eigenspectrum of $\mathbf{H}$ may lead to poor convergence rates, since eigenvalues far from each other may be grouped together, leading to a very large $\gamma_s$.

**Remark 2.** *To achieve good convergence rates, one should have some access to the eigenspectrum of the Hessian, in order to group together the indices of eigenvalues having the same order of magnitude.*

## F.1 PROOF OF THEOREM 1

*Proof.* We have:

$$\mathcal{L}(\boldsymbol{\theta}_{t+1}) = \frac{1}{2}\boldsymbol{\theta}_{t+1}^T \mathbf{H}\boldsymbol{\theta}_{t+1}$$

$$= \frac{1}{2}(\boldsymbol{\theta}_t - \mathbf{G}_t\mathbf{I}_{P:S}\boldsymbol{\eta}_t)^T \mathbf{H}(\boldsymbol{\theta}_t - \mathbf{G}_t\mathbf{I}_{P:S}\boldsymbol{\eta}_t)$$

$$= \mathcal{L}(\boldsymbol{\theta}_t) - \boldsymbol{\theta}_t^T \mathbf{H}\mathbf{G}_t\mathbf{I}_{P:S}\boldsymbol{\eta}_t + \frac{1}{2}\boldsymbol{\eta}_t^T \mathbf{I}_{S:P}\mathbf{G}_t\mathbf{H}\mathbf{G}_t\mathbf{I}_{P:S}\boldsymbol{\eta}_t$$

$$= \mathcal{L}(\boldsymbol{\theta}_t) - \boldsymbol{\theta}_t^T \mathbf{H}\mathbf{G}_t\mathbf{I}_{P:S}\boldsymbol{\eta}_t + \frac{1}{2}\boldsymbol{\eta}_t^T \bar{\mathbf{H}}_t\boldsymbol{\eta}_t$$

$$= \mathcal{L}(\boldsymbol{\theta}_t) - \mathbf{g}_t^T \mathbf{G}_t\mathbf{I}_{P:S}\bar{\mathbf{H}}_t^{-1}\bar{\mathbf{g}}_t + \frac{1}{2}\bar{\mathbf{g}}_t\bar{\mathbf{H}}_t^{-1}\bar{\mathbf{g}}_t$$

$$= \mathcal{L}(\boldsymbol{\theta}_t) - \frac{1}{2}\bar{\mathbf{g}}_t^T \bar{\mathbf{H}}_t^{-1}\bar{\mathbf{g}}_t.$$

Now, we study $\Delta = -\frac{1}{2}\bar{\mathbf{g}}^T \bar{\mathbf{H}}^{-1}\bar{\mathbf{g}}$. We omit the time $t$ for the sake of readability.

We can write $\mathbf{g}$ as a block vector: $\mathbf{g} = (\mathbf{g}_1, \cdots, \mathbf{g}_S)$, where $\mathbf{g}_s \in \mathbb{R}^{|\mathcal{I}_s|}$ for all $1 \leq s \leq S$. Thus, since $\mathbf{H}$ is block-diagonal:

$$\bar{\mathbf{H}} = \text{Diag}(\mathbf{g}_s^T \mathbf{H}_s \mathbf{g}_s : s \in \{1, \cdots, S\}),$$

$$\bar{\mathbf{H}}^{-1} = \text{Diag}((\mathbf{g}_s^T \mathbf{H}_s \mathbf{g}_s)^{-1} : s \in \{1, \cdots, S\}).$$

Also, $\bar{\mathbf{g}}_s = \mathbf{g}_s^T \mathbf{g}_s$, then:

$$\Delta = -\frac{1}{2}\sum_{s=1}^{S} \frac{(\mathbf{g}_s^T \mathbf{g}_s)^2}{\mathbf{g}_s^T \mathbf{H}_s \mathbf{g}_s}$$

$$= -\frac{1}{2}\sum_{s=1}^{S} \frac{(\mathbf{g}_s^T \mathbf{g}_s)^2(\mathbf{g}_s^T \mathbf{H}_s^{-1}\mathbf{g}_s)}{(\mathbf{g}_s^T \mathbf{H}_s \mathbf{g}_s)(\mathbf{g}_s^T \mathbf{H}_s^{-1}\mathbf{g}_s)}.$$

By Kantorovich's inequality, we have:

$$\Delta \leq -\frac{1}{2}\sum_{s=1}^{S} \frac{\mathbf{g}_s^T \mathbf{H}_s^{-1}\mathbf{g}_s}{\frac{1}{4}\left(\frac{a_s}{A_s} + \frac{A_s}{a_s} + 2\right)}$$

$$\leq -2\sum_{s=1}^{S} \frac{(\mathbf{g}_s^T \mathbf{H}_s^{-1}\mathbf{g}_s)A_s a_s}{(A_s + a_s)^2}.$$

Thus:

$$\Delta \leq -\min\left(\frac{2A_s a_s}{(A_s + a_s)^2}\right)\sum_{s=1}^{S} \mathbf{g}_s^T \mathbf{H}_s^{-1}\mathbf{g}_s$$

$$\leq -\min\left(\frac{2A_s a_s}{(A_s + a_s)^2}\right)\boldsymbol{\theta}^T \mathbf{H}\boldsymbol{\theta}.$$

Finally, when dividing by $\mathcal{L}(\boldsymbol{\theta}_t) = \frac{1}{2}\boldsymbol{\theta}^T \mathbf{H}\boldsymbol{\theta}$, we have:

$$\frac{\mathcal{L}(\boldsymbol{\theta}_{t+1})}{\mathcal{L}(\boldsymbol{\theta}_t)} - 1 \leq -\min\left(\frac{4A_s a_s}{(A_s + a_s)^2}\right)$$

$$\frac{\mathcal{L}(\boldsymbol{\theta}_{t+1})}{\mathcal{L}(\boldsymbol{\theta}_t)} \leq \max\left(\frac{(A_s - a_s)^2}{(A_s + a_s)^2}\right)$$

Besides, this rate is optimal, since it is possible to build $\boldsymbol{\theta}_t$ such that:

$$\frac{\mathcal{L}(\boldsymbol{\theta}_{t+1})}{\mathcal{L}(\boldsymbol{\theta}_t)} = \max_s\left(\frac{(A_s - a_s)^2}{(A_s + a_s)^2}\right).$$

To do so, let $s_0 \in \arg\max_s \left( \frac{(A_s - a_s)^2}{(A_s + a_s)^2} \right)$. Let $\mathbf{g}_{\min}$ be an eigenvector of $\mathbf{H}$ associated to $a_{s_0}$ and $\mathbf{g}_{\max}$ be an eigenvector of $\mathbf{H}$ associated to $A_{s_0}$, orthogonal with $\|\mathbf{g}_{\min}\| = \|\mathbf{g}_{\max}\| = 1$. Also, let $\boldsymbol{\theta}_t = \mathbf{H}^{-1}(\mathbf{g}_{\min} + \mathbf{g}_{\max})$.

Thus:

$$\mathcal{L}(\boldsymbol{\theta}_{t+1}) - \mathcal{L}(\boldsymbol{\theta}_t) = -\frac{1}{2}\frac{(\mathbf{g}_{s_0}^T \mathbf{g}_{s_0})^2}{\mathbf{g}_{s_0}^T \mathbf{H}_{s_0} \mathbf{g}_{s_0}} = -\frac{1}{2}\frac{2}{A_{s_0} + a_{s_0}}$$

Finally:

$$\frac{\mathcal{L}(\boldsymbol{\theta}_{t+1})}{\mathcal{L}(\boldsymbol{\theta}_t)} = 1 - \frac{1}{2}\frac{2}{A_{s_0} + a_{s_0}}\frac{1}{\frac{1}{2}\mathbf{g}_{s_0}^T \mathbf{H}_{s_0}^{-1} \mathbf{g}_{s_0}}$$

$$= 1 - \frac{2}{A_{s_0} + a_{s_0}}\frac{1}{A_{s_0}^{-1} + a_{s_0}^{-1}}$$

$$= 1 - \frac{2 A_{s_0} a_{s_0}}{(A_{s_0} + a_{s_0})^2}$$

$$= \frac{(A_{s_0} - a_{s_0})^2}{(A_{s_0} + a_{s_0})^2}$$

$\square$

## G EXPERIMENTAL DETAILS

**Practical implementation.** To implement the method proposed in Section 4, we propose Algorithm 4. The key function is compute_lr$(\lambda_{\text{int}}; \mathcal{L}, \boldsymbol{\theta}, \tilde{Z}, \mathbf{u})$, which returns a solution $\boldsymbol{\eta}_*$ of:

$$\boldsymbol{\eta}_* = \left( \bar{\mathbf{H}} + \frac{\lambda_{\text{int}}}{2}\|\mathbf{D}\boldsymbol{\eta}_*\|\mathbf{D}^2 \right)^{-1} \bar{\mathbf{g}},$$

$$\text{with:} \quad \bar{\mathbf{H}} := \mathbf{I}_{S:P}\text{Diag}(\mathbf{u})\frac{\mathrm{d}^2 \mathcal{L}}{\mathrm{d}\boldsymbol{\theta}^2}(\boldsymbol{\theta}, \tilde{Z})\text{Diag}(\mathbf{u})\mathbf{I}_{P:S},$$

$$\bar{\mathbf{g}} := \mathbf{I}_{S:P}\text{Diag}(\mathbf{u})\mathbf{g}_t,$$

$$\mathbf{D} := \text{Diag}\left( \left( \left| \mathbf{D}_{\boldsymbol{\theta}}^{(3)}(\mathbf{u}) \right|_{iii}^{1/3} \right)_{1 \leq i \leq S} \right).$$

"momentum$(\mu, \mathbf{x}, \tilde{\mathbf{x}})$" returns $\mathbf{x}$ if $\tilde{\mathbf{x}}$ is undefined, else $\mu\tilde{\mathbf{x}} + (1-\mu)\mathbf{x}$. "schedule$(\tau_{\text{sch}}, p_{\text{sch}}, f_{\text{sch}}; \cdots)$" corresponds to torch.optim.lr_scheduler.ReduceLROnPlateau called every $\tau_{\text{sch}}$ with patience $p_{\text{sch}}$ and factor $f_{\text{sch}}$, in order to reduce the step size $\lambda_t$ when the loss attains a plateau.[6] The samplers $\mathcal{D}_g$ and $\mathcal{D}_{\text{newt}}$ are respectively used to compute the gradients $\mathbf{g}_t$ and $(\bar{\mathbf{H}}, \bar{\mathbf{g}})$ used in "compute_lr".

The hyperparameters are: the initial step size $\lambda_1$, the momentum $\mu_g$ on the gradients $\mathbf{g}_t$, the minibatch size $B$ to sample the $\tilde{Z}$ (used to compute $\bar{\mathbf{g}}$, $\bar{\mathbf{H}}$ and $\mathbf{D}$), the number of steps $\tau$ between each call of compute_lr, the momentum $\mu_\eta$ on the learning rates $\boldsymbol{\eta}_t$, the internal damping $\lambda_{\text{int}}$, and the parameters of the scheduler $\tau_{\text{sch}}, p_{\text{sch}}, f_{\text{sch}}$.

**Explanation.** The "momentum" functions are used to deal with the stochastic part of the training process, since our method has not been designed to be robust against noise. The period $\tau$ is usually strictly greater than 1, in order to avoid calling "compute_lr" at every step, which would be costly. The minibatch size $B$ should be large enough to reduce noise in the estimation of $\boldsymbol{\eta}_*$. If we denote by $B_g$ the size of the minibatches in $\mathcal{D}_g$, then we recommend the following setup: $\tau = \frac{B}{B_g} = \frac{1}{1-\mu_g}$. That way, we ensure that the training data are sampled from $\mathcal{D}_g$ and $\mathcal{D}_{\text{newt}}$ at the same rate, and that $\tilde{\mathbf{g}}_t$ memorizes the preceding gradients $\mathbf{g}_t$ for $\tau$ steps. Besides, we have to take the positive part $(\boldsymbol{\eta}_t)_+$ of $\boldsymbol{\eta}_t$ in order to avoid negative learning rates.

---

[6]See torch.optim.lr_scheduler.ReduceLROnPlateau.

---

**Algorithm 4** Complete implementation of the second-order optimization method described in Sec. 4. $\lambda_1$ and $\lambda_{\text{int}}$ are the only hyperparameter to be tuned across the experiments, the others are fixed.

---

Hyperparams: $\lambda_1, \mu_g, B_g, B, \tau, \mu_\eta, \lambda_{\text{int}}, \tau_{\text{sch}}, p_{\text{sch}}, f_{\text{sch}}$
$\mathcal{D}_{\text{g}} \leftarrow$ sampler of minibatches of size $B_g$
$\mathcal{D}_{\text{newt}} \leftarrow$ sampler of minibatches of size $B$
**for** $t \in [1, T]$ **do**
    $Z_t := (X_t, Y_t) \sim \mathcal{D}_{\text{g}}$                                                       (sample minibatch)
    $\mathcal{L}_t \leftarrow \mathcal{L}(\boldsymbol{\theta}_t, Z_t)$                                                            (forward pass)
    $\mathbf{g}_t \leftarrow \frac{\mathrm{d}\mathcal{L}}{\mathrm{d}\boldsymbol{\theta}}(\boldsymbol{\theta}_t, Z_t)$                                                   (backward pass)
    $\tilde{\mathbf{g}}_t \leftarrow \text{momentum}(\mu_g; \mathbf{g}_t, \tilde{\mathbf{g}}_{t-1})$
    **if** $t \% \tau == 0$ **then**
        sample $\tilde{Z}_t \sim \mathcal{D}_{\text{newt}}$
        $\boldsymbol{\eta}_t \leftarrow \text{compute\_lr}(\lambda_{\text{int}}; \mathcal{L}, \boldsymbol{\theta}_t, \tilde{Z}_t, \tilde{\mathbf{g}}_t)$
        $\tilde{\boldsymbol{\eta}}_t \leftarrow \text{momentum}(\mu_\eta; (\boldsymbol{\eta}_t)_+, \tilde{\boldsymbol{\eta}}_{t-1})$
    **end if**
    $\boldsymbol{\theta}_{t+1} \leftarrow \boldsymbol{\theta}_t - \lambda_t \text{Diag}(\tilde{\mathbf{g}}_t) \mathbf{I}_{P:S} \tilde{\boldsymbol{\eta}}_t$                                (training step)
    $\lambda_{t+1} \leftarrow \text{schedule}(\tau_{\text{sch}}, p_{\text{sch}}, f_{\text{sch}}; t, \mathcal{L}_t, \lambda_t)$
**end for**

---

**Experimental setup.** We provide in Table 2 the hyperparameters fixed for all the experiments. In Table 3, we report the results of the grid-search for the hyperparameters of the 3 tested optimization methods.

Table 2: Hyperparameters fixed in all the series of experiments. $N_e$ is the number of training steps per epoch.

| $\mu_g$ | $B_g$ | $B$ | $\tau$ | $\mu_\eta$ | $\tau_{\text{sch}}$ | $p_{\text{sch}}$ | $f_{\text{sch}}$ |
|---------|-------|-----|--------|------------|---------------------|------------------|------------------|
| 0.9 | $10^2$ | $10^3$ | 10 | 0.5 | $N_e$ | 5 | 0.5 |

Table 3: Hyperparameters tuned for each series of experiments. $\eta$: learning rate, $\lambda_1$: initial step size.

|  |  | MLP | LeNet | VGG-11' | BigMLP |
|------|------|------|------|------|------|
| Adam | $\eta$ | $3 \cdot 10^{-4}$ | $3 \cdot 10^{-4}$ | $10^{-5}$ | $10^{-5}$ |
| KFAC | $\eta$ | $10^{-4}$ | $10^{-4}$ | $3 \cdot 10^{-4}$ | $10^{-5}$ |
|  | $\lambda$ | $10^{-2}$ | $3 \cdot 10^{-2}$ | $3 \cdot 10^{-2}$ | $10^{-2}$ |
| Ours | $\lambda_1$ | $10^{-1}$ | $3 \cdot 10^{-1}$ | $3 \cdot 10^{-1}$ | $10^{-1}$ |
|  | $\lambda_{\text{int}}$ | 10 | 3 | 3 | 10 |

**K-FAC update periods.** In accordance with the K-FAC packages, we have chosen to increase the update period of the pre-conditioner to reduce the training time. Specifically, we have chosen to perform a covariance update every 10 steps, and the inversion of the Fisher matrix every 100 steps:

- with tensorflow/kfac: use PeriodicInvCovUpdateKfacOpt with: cov_update_every = 10 and invert_every = 100;

- with alecwangcq/KFAC-Pytorch: use KFACOptimizer with: TCov = 10 and TInv = 100.

## H VERY DEEP MULTILAYER PERCEPTRON

**Grouping the layers.** In addition to the neural networks considered in Section 5, we have also tested "VBigMLP", a very deep multilayer perceptron with 100 layers of size 1024 trained on CIFAR-10.

Instead of considering $S = 2L = 200$ groups of parameters, we split the sequence of layers of VBigMLP into 5 chunks. Then, each chunk is divided into 2 parts, one containing the weight tensors, and the other the bias vectors. Finally, we have $S = 10$ subsets of parameters, grouped by role (weight/bias) and by position inside the network.

**Experimental results.** We show in Figure 3a the matrices $\bar{\mathbf{H}}$ and $\bar{\mathbf{H}}^{-1}$ at different stages of training. At initialization, even if the neural network is very deep, we observe that all the chunks of the network interact together, even the first one with the last one. However, after several training steps, the long-range interactions seem to disappear. Incidentally, the matrices become tridiagonal, which ties in with the block-tridiagonal approximation of the inverse of the Hessian done by Martens & Grosse (2015).

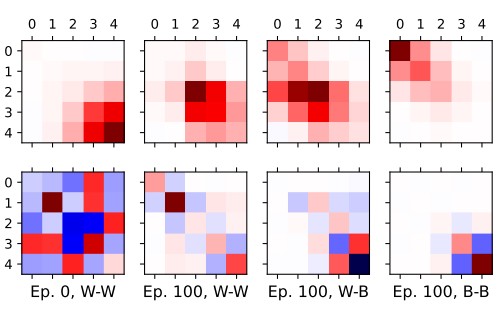

(a) Submatrices of $\bar{\mathbf{H}}$ (first row) and $\bar{\mathbf{H}}^{-1}$ (second row), at initialization and before the 100th epoch.

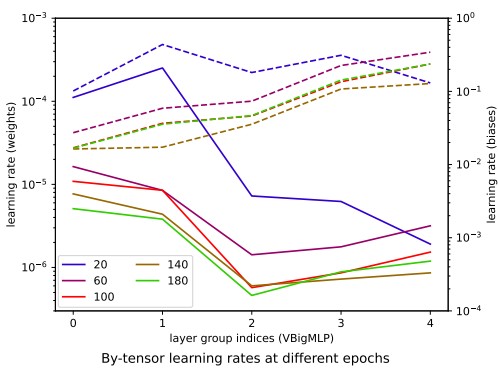

(b) Learning rates $\eta_*$ computed according to 10, specific to each subset of parameters.

Figure 3: Matrices $\bar{\mathbf{H}}$ and $\bar{\mathbf{H}}^{-1}$ and per-subset-of-parameters learning rates obtained with VBigMLP. Legend for the figure on the right: solid lines: weights; dotted lines: biases. For each epoch $k \in \{20, 60, 100, 140, 180\}$, the reported value has been averaged over the epochs $[k - 20, k + 19]$ to remove the noise.

In Figure 3b, we observe the evolution of the learning rates $\eta_*$ computed according to 10. First, there are all decreasing during training. Second, the biases in the last layers of the network seem to need larger learning rates than biases in the first layers. Third, the learning rate computed for the weights of the first chunk of layers is smaller than the others.

Finally, the training curves in Figure 4a indicate that our method can be used to train very deep networks. In this setup, it is close to be competitive with Adam. Besides, we did not manage to tune the learning rate and the damping of K-FAC to make it work in this setup.

We have also plotted the evolution of the test loss and test accuracy during training (see Figure 4b). It is clear that Adam does not generalize at all, while our method attains a test accuracy around 35 % – 40 %.

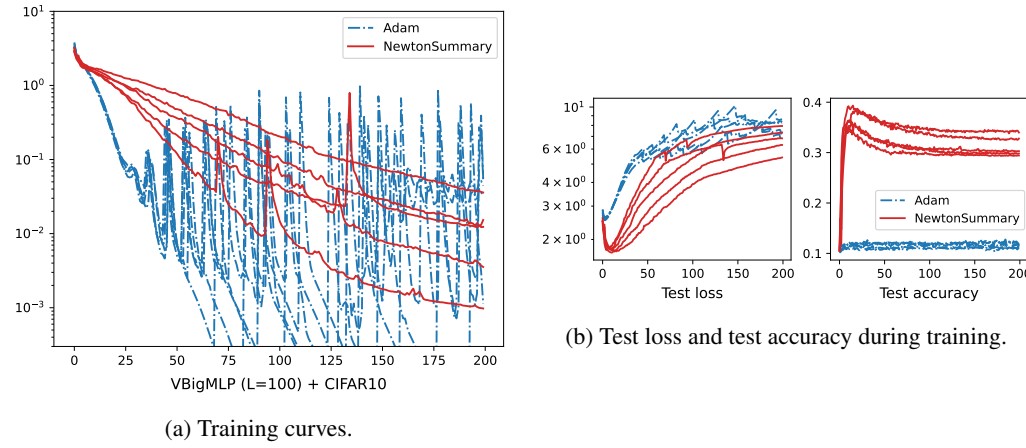

(a) Training curves.

(b) Test loss and test accuracy during training.

Figure 4: VBigMLP + CIFAR-10.

## I  TEST LOSS AND TEST ACCURACY

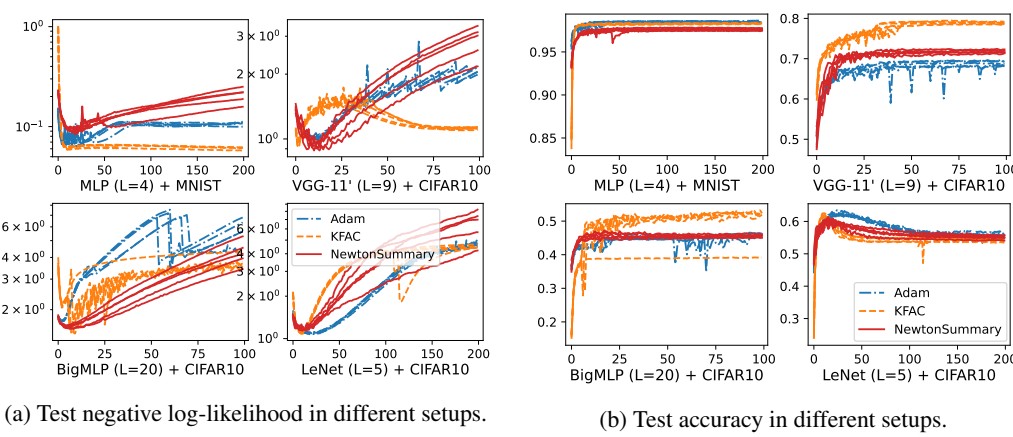

(a) Test negative log-likelihood in different setups.

(b) Test accuracy in different setups.

Figure 5: Test metrics in various setups.

In Figure 5a and Figure 5b, we have reported the test negative log-likelihood and the test accuracy of the same experiments as in Section 5.2 (Figure 2a).

Our method is competitive with Adam and K-FAC when comparing the test losses, except for the MLP trained on MNIST. In several cases, we observe a discrepancy between the test loss and the test accuracy: one method might be better than another according to the loss, but worse in terms of accuracy. In particular, the test loss of our method can achieve smaller test losses than the other methods, while its test accuracy remains slightly lower (BigMLP, VGG).

## J  CHOICE OF THE PARTITION

We have trained VGG-11' on CIFAR-10 using our method with different partition choices. In Table 4, we report the final training losses, the training time (wall-time), and the maximum memory usage.

Not surprisingly, the finer the partition, the better the results. However, this comes at a cost: training with finer partitions takes more time. We also observe that memory usage tends to decrease as the partition becomes finer.

The partitions we have tested are:

- trivial, $S = 1$: all the tensors are grouped together;

- weights-biases, $S = 2$: all the weights are grouped together, and all the biases too;
- blocks-$k$, $S = 2k + 2$: the sequence of convolutional layers is split into $k$ consecutive blocks, and each one is split in two (weights + biases); weights and biases of the final fully-connected layer are considered separately (hence the "+2" in $S$);
- alternate-$k$, $S = 2k + 2$: the convolutional layer $l$ is put in the $\tilde{s}$-block if $l\%k = \tilde{s}$; then, each block is split in two (weights + biases); weights and biases of the final fully-connected layer are considered separately (hence the "+2" in $S$);
- canonical, $S = \#$tensors: each tensor is considered separately.

Table 4: Influence of the choice of the partition when training VGG-11' on CIFAR-10.

| partition | train NLL | time (s) | mem. (Go) |
|---|---|---|---|
| trivial | $8.12 \cdot 10^{-1}$ | 2 512 | 2.49 |
| weights-biases | $7.64 \cdot 10^{-1}$ | 2 855 | 2.49 |
| blocks-2 | $5.94 \cdot 10^{-1}$ | 3 182 | 2.38 |
| alternate-2 | $5.70 \cdot 10^{-1}$ | 3 422 | 2.08 |
| blocks-4 | $1.50 \cdot 10^{-2}$ | 3 674 | 1.97 |
| alternate-4 | $5.37 \cdot 10^{-2}$ | 4 180 | 1.91 |
| canonical | $3.05 \cdot 10^{-4}$ | 4 612 | 1.88 |

## K    MEASURING THE IMPORTANCE OF INTERACTIONS BETWEEN LAYERS

**Diagonal approximation of Method 1.**    Throughout this paper, we have emphasized the importance of considering the interactions between layers when training a neural network. In fact, Method 1 allows the user to keep track of them at a reasonable computational cost. But is it useful to take these interactions into account?

If the computational cost is really an issue, one can compute only the diagonal coefficients of $\bar{\mathbf{H}}$ and set the off-diagonal coefficients to zero. Let $\bar{\mathbf{H}}^0$ be this *diagonal approximation* of $\bar{\mathbf{H}}$:

$$\bar{\mathbf{H}}^0 := \mathrm{Diag}((\bar{h}_{ii})_{1 \le i \le S}),$$

where $(\bar{h}_{ii})_{1 \le i \le S}$ are the diagonal coefficients of $\bar{\mathbf{H}}$.

Then, we call the *diagonal approximation* of Method 1, Method 1 where $\bar{\mathbf{H}}$ has been replaced by $\bar{\mathbf{H}}^0$.

**Experiments.**    We have tested Method 1 with the hyperparameters we have used in Section 5.2 and its diagonal approximation with a grid of hyperparameters $\lambda_1$ and $\lambda_{\mathrm{int}}$. The results are shown in Figure 6. Note that the configuration $\lambda_1 = 1$ was tested with VGG11', but resulted in instantaneous divergence, so we have not plotted the corresponding training curves.

According to Figure 6, the diagonal approximation of Method 1 performs worse or is more unstable than 1. Therefore, when training LeNet or VGG11' with CIFAR10, it is better to keep the off-diagonal coefficients of $\bar{\mathbf{H}}$.

In short, one should worry about the interactions between layers.

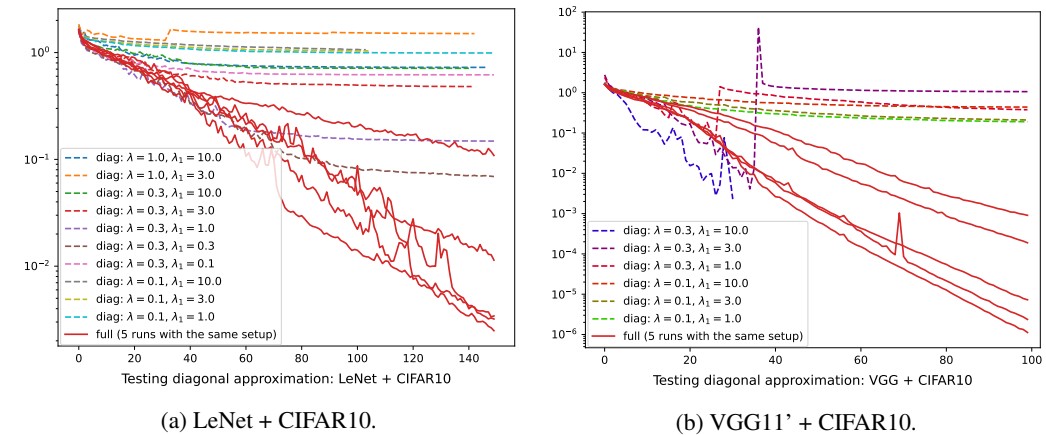

(a) LeNet + CIFAR10.   (b) VGG11' + CIFAR10.

Figure 6: Training curves: Method 1 (solid lines) versus its diagonal approximation (dotted lines) with various hyperparameters.

## L  HIGHER-ORDER DERIVATIVES OF A MULTIVARIATE FUNCTION

In this section, we recall formally the definition of higher-order derivatives of a multivariate function, following Dieudonné (1960).

### L.1  DEFINITIONS

Let $\mathcal{L}(E, F)$ be the space of linear maps from $E$ to $F$ and $\mathcal{L}_d(E, F)$ be the space of $d$-linear maps from $E \times \cdots \times E$ to $F$. For instance, the space of linear forms on $\mathbb{R}^d$ is denoted by $\mathcal{L}(\mathbb{R}^P, \mathbb{R})$, and the space of 3-linear forms on $\mathbb{R}^P \times \mathbb{R}^P \times \mathbb{R}^P$ is denoted by $\mathcal{L}_3(\mathbb{R}^P, \mathbb{R})$.

Let $f$ be a smooth multivariate function from $\mathbb{R}^P$ to $\mathbb{R}$:

$$f : \mathbb{R}^P \to \mathbb{R}. \tag{47}$$

**Differential of order 1.**   The differential of $f$ at a point $\boldsymbol{\theta} \in \mathbb{R}^P$ is the only linear form $T_f(\boldsymbol{\theta}) \in \mathcal{L}(\mathbb{R}^P, \mathbb{R})$ such that:

$$\lim_{\mathbf{u} \to 0} \frac{f(\boldsymbol{\theta} + \mathbf{u}) - f(\boldsymbol{\theta}) - T_f(\boldsymbol{\theta})(\mathbf{u})}{\|\mathbf{u}\|^2} = 0. \tag{48}$$

Since $T_f(\boldsymbol{\theta})$ is a linear form, there exists a vector $\mathbf{g} \in \mathbb{R}^P$ such that:

$$T_f(\boldsymbol{\theta})(\mathbf{u}) = \mathbf{g}^T \mathbf{u}. \tag{49}$$

The vector $\mathbf{g}$ is nothing other than the *gradient* of $f$ at $\boldsymbol{\theta}$, and $T_f(\boldsymbol{\theta})$ is the *differential* of $f$ at $\boldsymbol{\theta}$, that we denote by $\frac{\mathrm{d}f}{\mathrm{d}\boldsymbol{\theta}}(\boldsymbol{\theta})$ in the main text.

In addition, there is a relationship between the coordinates $g_i$ of the gradient $\mathbf{g} = (g_1, \cdots, g_P)$ and the differential $\frac{\mathrm{d}f}{\mathrm{d}\boldsymbol{\theta}}(\boldsymbol{\theta})$:

$$\forall i \in \{1, \cdots, P\}, \quad g_i = \frac{\mathrm{d}f}{\mathrm{d}\boldsymbol{\theta}}(\boldsymbol{\theta})(\mathbf{e}_i), \tag{50}$$

where $\mathbf{e}_i = (0, \cdots, 0, 1, 0, \cdots, 0) \in \mathbb{R}^P$ is the $i$-th vector of the canonical basis (in other words, $\mathbf{e}_i$ is the one-hot representation of the integer $i$).

And, of course, the $g_i$ can be calculated by using the partial derivatives:

$$\forall i \in \{1, \cdots, P\}, \quad g_i = \frac{\partial f}{\partial \theta_i}(\boldsymbol{\theta}). \tag{51}$$

**Differential of order $d$.** We suppose that the differential of order $d-1$ of $f$ at $\boldsymbol{\theta}$ is well-defined and is a $(d-1)$-linear form on $\mathbb{R}^P$. We denote it by:

$$\frac{\mathrm{d}^{d-1}f}{\mathrm{d}\boldsymbol{\theta}^{d-1}}(\boldsymbol{\theta}) \in \mathcal{L}_{d-1}(\mathbb{R}^P, \mathbb{R}). \tag{52}$$

Thus, one can apply $\frac{\mathrm{d}^{d-1}f}{\mathrm{d}\boldsymbol{\theta}^{d-1}}(\boldsymbol{\theta})$ to a sequence of vectors $(\mathbf{u}^1, \cdots, \mathbf{u}^{d-1})$. We can write:

$$\frac{\mathrm{d}^{d-1}f}{\mathrm{d}\boldsymbol{\theta}^{d-1}}(\boldsymbol{\theta}) : \mathbb{R}^P \times \cdots \times \mathbb{R}^P \to \mathbb{R}$$

$$(\mathbf{u}^1, \cdots, \mathbf{u}^{d-1}) \mapsto \frac{\mathrm{d}^{d-1}f}{\mathrm{d}\boldsymbol{\theta}^{d-1}}(\boldsymbol{\theta})(\mathbf{u}^1, \cdots, \mathbf{u}^{d-1}). \tag{53}$$

Now, given a sequence of vectors $(\mathbf{u}^1, \cdots, \mathbf{u}^{d-1})$, let us define $g(\cdot)[\mathbf{u}^1, \cdots, \mathbf{u}^{d-1}] : \mathbb{R}^P \to \mathbb{R}$ such that:

$$g(\boldsymbol{\theta})[\mathbf{u}^1, \cdots, \mathbf{u}^{d-1}] = \frac{\mathrm{d}^{d-1}f}{\mathrm{d}\boldsymbol{\theta}^{d-1}}(\boldsymbol{\theta})(\mathbf{u}^1, \cdots, \mathbf{u}^{d-1}). \tag{54}$$

So, $g(\cdot)[\mathbf{u}^1, \cdots, \mathbf{u}^{d-1}]$ is a function from $\mathbb{R}^P$ to $\mathbb{R}$, and $g(\boldsymbol{\theta})[\cdot] \in \mathcal{L}_{d-1}(\mathbb{R}^P, \mathbb{R})$.

As a smooth function from $\mathbb{R}^P$ to $\mathbb{R}$, one can compute the differential of $g(\cdot)[\mathbf{u}^1, \cdots, \mathbf{u}^{d-1}]$ at $\boldsymbol{\theta}$, that is a linear form:

$$\frac{\mathrm{d}g}{\mathrm{d}\boldsymbol{\theta}}(\boldsymbol{\theta})[\mathbf{u}^1, \cdots, \mathbf{u}^{d-1}] : \mathbb{R}^P \to \mathbb{R}$$

$$\mathbf{u}^d \mapsto h(\boldsymbol{\theta})(\mathbf{u}^d) = \frac{\mathrm{d}g}{\mathrm{d}\boldsymbol{\theta}}(\boldsymbol{\theta})[\mathbf{u}^1, \cdots, \mathbf{u}^{d-1}](\mathbf{u}^d). \tag{55}$$

We change the notation slightly by setting:

$$\frac{\mathrm{d}g}{\mathrm{d}\boldsymbol{\theta}}(\boldsymbol{\theta})[\mathbf{u}^1, \cdots, \mathbf{u}^{d-1}, \mathbf{u}^d] := \frac{\mathrm{d}g}{\mathrm{d}\boldsymbol{\theta}}(\boldsymbol{\theta})[\mathbf{u}^1, \cdots, \mathbf{u}^{d-1}](\mathbf{u}^d). \tag{56}$$

With this notation, it can be proven that $\frac{\mathrm{d}g}{\mathrm{d}\boldsymbol{\theta}}(\boldsymbol{\theta})[\cdot]$ is a $d$-linear form (it belongs to $\mathcal{L}_d(\mathbb{R}^P, \mathbb{R})$). Finally, by definition of $g$:

$$\frac{\mathrm{d}g}{\mathrm{d}\boldsymbol{\theta}}(\boldsymbol{\theta})[\mathbf{u}^1, \cdots, \mathbf{u}^{d-1}, \mathbf{u}^d] = \frac{\mathrm{d}}{\mathrm{d}\boldsymbol{\theta}}\left(\frac{\mathrm{d}^{d-1}f}{\mathrm{d}\boldsymbol{\theta}^{d-1}}(\boldsymbol{\theta})(\mathbf{u}^1, \cdots, \mathbf{u}^{d-1})\right)(\mathbf{u}^d), \tag{57}$$

that we denote by:

$$\frac{\mathrm{d}^d f}{\mathrm{d}\boldsymbol{\theta}^d}(\boldsymbol{\theta})[\mathbf{u}^1, \cdots, \mathbf{u}^{d-1}, \mathbf{u}^d]. \tag{58}$$

So, $\frac{\mathrm{d}^d f}{\mathrm{d}\boldsymbol{\theta}^d}(\boldsymbol{\theta}) \in \mathcal{L}_d(\mathbb{R}^P, \mathbb{R})$.

Like the order-1 differential, the order-$d$ differential can be represented by a tensor. For instance, a canonical representation of $\frac{\mathrm{d}^d f}{\mathrm{d}\boldsymbol{\theta}^d}(\boldsymbol{\theta})$ is $\mathbf{T} \in \mathbb{R}^{P^d}$ with:

$$T_{i_1, \cdots, i_d} = \frac{\mathrm{d}^d f}{\mathrm{d}\boldsymbol{\theta}^d}(\boldsymbol{\theta})[\mathbf{e}_{i_1}, \cdots, \mathbf{e}_{i_d}] \in \mathbb{R}, \tag{59}$$

where $T_{i_1, \cdots, i_d}$ is the value located at index $(i_1, \cdots, i_d)$ in $\mathbf{T}$.

We can also define $\mathbf{T}$ with partial derivatives:

$$T_{i_1, \cdots, i_d} = \frac{\partial^d f}{\partial \theta_{i_1} \cdots \partial \theta_{i_d}}(\boldsymbol{\theta}) \in \mathbb{R}. \tag{60}$$

**Example with $d=2$.** With $d=2$, the tensor $\mathbf{T}$ representing the order-2 differential is the Hessian matrix. So, $\mathbf{T} \in \mathbb{R}^{P^2}$ with:

$$T_{ij} = \frac{\mathrm{d}^2 f}{\mathrm{d}\boldsymbol{\theta}^2}(\boldsymbol{\theta})[\mathbf{e}_i, \mathbf{e}_j] = \frac{\partial^2 f}{\partial \theta_i \partial \theta_j}(\boldsymbol{\theta}). \tag{61}$$

## L.2 PARTIAL DERIVATIVES WITH RESPECT TO VECTORS

We also need to define formally the following notation, used in Section 3:

$$\frac{\partial^d f}{\partial \mathbf{T}^{i_1} \cdots \partial \mathbf{T}^{i_d}}(\boldsymbol{\theta}). \tag{62}$$

Without loss of generality, we only consider the case where the $\mathbf{T}^i$ are vectors (and not higher-order tensors).

**Representation of $\theta$ as a sequence of vectors.** We consider that the argument $\boldsymbol{\theta} \in \mathbb{R}^P$ of the function $f$ can be represented as a sequence of $S$ vectors. For instance:

$$\boldsymbol{\theta} = (\theta_1, \cdots, \theta_P) \cong ((\theta_1, \theta_3, \theta_5, \cdots), (\theta_2, \theta_4, \theta_6, \cdots)), \tag{63}$$

$$\text{or } \boldsymbol{\theta} = (\theta_1, \cdots, \theta_P) \cong ((\theta_1, \theta_2, \cdots, \theta_{P_1}), (\theta_{P_1+1}, \cdots, \theta_{P_1+P_2}),$$
$$(\theta_{P_1+P_2+1}, \cdots, \theta_{P_1+P_2+P_3}), \cdots), \tag{64}$$

etc.,

where $P_1, P_2, \cdots, P_S$ are integers such that $P_1 + \cdots + P_S = P$, and "$\cong$" means "is represented by". It is essential that each $\theta_i$ appears exactly once in the right-hand side of the equations above.

Without loss of generality, $\boldsymbol{\theta}$ can be represented by a sequence of $S$ vectors with defined sizes $(P_1, \cdots, P_S)$:

$$\boldsymbol{\theta} \cong (\mathbf{T}^1, \mathbf{T}^2, \cdots, \mathbf{T}^S) \in \mathbb{R}^{P_1} \times \mathbb{R}^{P_2} \times \cdots \times \mathbb{R}^{P_S}. \tag{65}$$

**Single partial derivative.** Let $\mathbf{u} \in \mathbb{R}^P$ be a vector. Just as for $\boldsymbol{\theta}$, we represent $\mathbf{u}$ by a sequence of vectors:

$$\mathbf{u} \cong (\mathbf{U}^1, \mathbf{U}^2, \cdots, \mathbf{U}^S) \in \mathbb{R}^{P_1} \times \mathbb{R}^{P_2} \times \cdots \times \mathbb{R}^{P_S}. \tag{66}$$

To be more specific, if $\mathbf{T}^i$ contains $(\theta_1, \theta_3, \theta_6)$, then $\mathbf{U}^i$ contains $(u_1, u_3, u_6)$.

Then, we can define $\frac{\partial f}{\partial \mathbf{T}^i}(\boldsymbol{\theta})$ as a linear form belonging to $\mathcal{L}(\mathbb{R}^{P_i}, \mathbb{R})$ with the following property:

$$\frac{\partial f}{\partial \mathbf{T}^i}(\boldsymbol{\theta}) : \mathbb{R}^{P_i} \to \mathbb{R} \tag{67}$$

$$\mathbf{U}^i \mapsto \frac{\partial f}{\partial \mathbf{T}^i}(\boldsymbol{\theta})[\mathbf{U}^i] = \sum_{k=1}^{P_i} \frac{\partial f}{\partial T_k^i}(\boldsymbol{\theta}) U_k^i, \tag{68}$$

where $T_k^i$ is the $k$-th coordinate of $\mathbf{T}^i$ and $U_k^i$ is the $k$-th coordinate of $\mathbf{U}^i$. To be more specific, if $T_k^i$ represents $\theta_q$, then $\frac{\partial f}{\partial T_k^i}(\boldsymbol{\theta}) = \frac{\partial f}{\partial \theta_q}(\boldsymbol{\theta})$.

**Multiple partial derivatives.** We can define $\frac{\partial^d f}{\partial \mathbf{T}^{i_1} \cdots \partial \mathbf{T}^{i_d}}(\boldsymbol{\theta})$ as a $d$-linear form belonging to $\mathcal{L}(\mathbb{R}^{P_{i_1}} \times \cdots \times \mathbb{R}^{P_{i_d}}, \mathbb{R})$ with the following property:

$$\frac{\partial^d f}{\partial \mathbf{T}^{i_1} \cdots \partial \mathbf{T}^{i_d}}(\boldsymbol{\theta}) : \mathbb{R}^{P_{i_1}} \times \cdots \times \mathbb{R}^{P_{i_d}} \to \mathbb{R}$$

$$(\mathbf{U}^{i_1}, \cdots, \mathbf{U}^{i_d}) \mapsto \frac{\partial^d f}{\partial \mathbf{T}^{i_1} \cdots \partial \mathbf{T}^{i_d}}(\boldsymbol{\theta})[\mathbf{U}^{i_1}, \cdots, \mathbf{U}^{i_d}]$$

$$= \sum_{k_1=1}^{P_1} \cdots \sum_{k_d=1}^{P_d} \frac{\partial f}{\partial T_{k_1}^{i_1} \cdots \partial T_{k_d}^{i_d}}(\boldsymbol{\theta}) U_{k_1}^{i_1} \cdots U_{k_d}^{i_d}. \tag{69}$$

## M COMPARISON OF TRAINING TIMES

In Table 5, we report the training times of 4 different neural networks with Adam, K-FAC and our method. Each value is the training time (wall-time) of the configuration in seconds, averaged over 5

runs. Note that MLP and LeNet were trained over 200 epochs, while BigMLP and VGG were trained over 100 epochs, which explains why the training times are larger for smaller networks.

For small networks (MLP, LeNet), the training times are very close with the different optimizers. However, we observe significant differences with large networks (BigMLP, VGG): compared to Adam, the training is 2 times longer with K-FAC and 3 times longer with our method.

Thus, the computational overhead of our method is either very small or not excessively large compared to K-FAC.

Table 5: Comparison of training times (in seconds) of different optimization techniques for the 4 main setups.

| Setup | Adam | K-FAC | Ours |
|---|---|---|---|
| MLP | 2848 | 2953 | 3315 |
| LeNet | 2944 | 3022 | 3369 |
| BigMLP | 1777 | 2989 | 4365 |
| VGG | 1696 | 3117 | 4613 |

