# OpenReview forum: "Gathering and Exploiting Higher-Order Information when Training Large Structured Models"
_ICLR.cc/2025/Conference — Submitted to ICLR 2025_

### Official Review · Reviewer_PhVC · 2024-11-03

**Soundness:** 2
**Presentation:** 2
**Contribution:** 2
**Rating:** 5
**Confidence:** 3

**Summary:**

The authors suggest a layer-wise partitioning of order $d$ derivatives, that transforms the $R^{p^d}$ tensor to a $R^{S^d}$ tensor which is computationally tractable even in deep networks.  The authors then leverage this partitioning to first compute empirically the Hessian for some deep neural networks, and subsequently suggest a second order method based on partitioning for optimization.

**Strengths:**

The paper has several strengths:

1) The interest in efficient computational schemes of higher order information for deep neural networks is significant, and improved methods of estimating the Hessian, as well as higher order terms could help improve interpretability and shed light on what neural networks learn during optimization.
2) The partitioning scheme is the main contribution of this paper, and seems to be novel as far as I know, with possible applications to many interesting avenues.

**Weaknesses:**

In spite of the strengths, the paper has some clear drawbacks in my opinion:

1) The paper is not written well enough, with a substantial lack of a literature survey on properties of Hessians in deep networks, as well as works on second order methods from recent years. In terms of the writing itself, all of the equations on page 5 are unnumbered making them hard to refer to, and the chosen notation for tensor contraction $A[ u, u... u]$ is not standard and never explained. It must be understood from context in the main text or the appendices.
2) The main contribution - the fast computation of lower complexity tensor containing similar information as the original tensor is not given in detail in the main text, and even in the appendix should be explained explicitly.
3) In general, I believe the focus of the paper is completely misguided. It seems clear that the suggestion of the second order optimization method is not superior (at least in its current form) to other simple gradient based methods, and should then not be the focus of the paper. Instead, if the paper focused on the partitioning/computation methods and then applied this to real-world or even particular solvable examples, extending Sec. 5.1, the paper would be much stronger.

**Questions:**

1) L43 - I'm not sure this is correct, assuming unlimited compute, and a perfectly known Hessian, even if the Hessian is singular you can still invert it on the nonsingular subspace, using SVD and find the pseudo-inverse.
2) L71 - Why do the authors need to regularize instead of computing just the pseudo inverse? is it inefficient? this should be stated explicitly
3) L76 - "so its preserves "
4) L218-220 seems wrong, the d-derivative is a map from $\mathbb{R}$ to $\mathbb{R}^{p^d}$ and not the other way around, even if the intention was a map from weight space to the operator output it should be from $\mathbb{R}^p$ to $\mathbb{R}^{p^d}$, so this is unclear to me. Additionally, the second term is unclear, $u$ are not defined at this point and the brackets $[.]$ are undefined as well. In the appendix it seems clear that the intention is tensor contraction between the previous term and the brackets but this is not standard.
5) L258 - "Therefore, the tensors $D_\theta^d(u)$ extract more information than the naive Taylor terms, while keeping a reasonable computational cost. " why is this statement obvious "therefore"? is the statement that regular Taylor terms lose the layer-wise structure of the network? I understand that the equality between Taylor and this decomposition is only obtained after tracing out the $s_i$ indices, but what is the intuition? it would be useful to show explicitly for a low $d$ derivative with a fixed number of parameters to illustrate the difference.
6) the authors don't comment on the compute time of their method compared to single gradient based method (Which seem to be better so far), it would seem like $t*p*S$ vs $t*p$, making it substantially slower for deep networks.
7) While the method provided in this text is not shown to be superior to standard algorithms, it might be interesting to consider the computation method in the context of sharpness aware minimization.

---

> ### Author Response · Authors · 2024-11-21
>
> We thank Reviewer PhVC for the review and the insightful remarks and suggestions. We also appreciate that the reviewer acknowledges our motivation and the relevance of our method.
>
> # Quality of writing
>
> We have made many corrections (highlighted in blue). We hope that the writing quality of the paper is now sufficient.
>
> The equations have been numbered.
>
> # Higher-order derivatives of a multivariate function
>
> The reviewer raises an important clarity issue, since it is crucial to understand the notation used in Section 3 to understand the rest of the paper.
>
> In short, we have dropped the "linear form" formalism and adopted fully the "tensor" formalism for the derivatives. For tensor contraction, to our knowledge, Einstein notation is the standard, but it is cumbersome, and we feel that it is too complicated for our use. So, we used a notation similar to [6, 7], based on [5] (see Chapter VIII.12), and detailed for instance in http://virtualmath1.stanford.edu/~conrad/diffgeomPage/handouts/taylor.pdf .
>
> Anyway, we have improved Section 3 and Appendix A, and we have added Appendix L to make our notation as clear as possible.
>
> # Use of pseudo-inverse in optimization
>
> We thank the reviewer for the suggestion. However, the pseudo-inverse has many drawbacks, which are not related to the computational cost. Notably, **after checking well-known reference books in optimization [1, 2, 3], computing the pseudo-inverse of the Hessian (or any other matrix) when it is singular is extremely uncommon.** We explain why in the following.
>
> During the practical optimization process (e.g. when using Newton's method), the matrices we need to invert are in fact invertible: it is very rare for an eigenvalue of the Hessian to be numerically zero. So, there is no need to use the pseudo-inverse: when a matrix is invertible, its inverse is equal to its pseudo-inverse.
>
> However, it is hard to deal with close-to-zero eigenvalues: it is technically possible to invert a matrix with close-to-zero eigenvalues, but it would result in a matrix with very large eigenvalues (leading to exploding numerical values, instabilities of the optimization process, etc.). The pseudo-inverse would be useless to solve this problem, since it is equal to the inverse (whether the eigenvalues are close to zero or not). Therefore, other techniques have been developed to overcome this difficulty (see [1, 2, 3]). A common one is regularization: instead of inverting $\mathbf{A}$, we invert $\mathbf{A}+\epsilon I_P$ (where $I_P$ is the identity matrix).
>
> We have built our own regularization on top of the one presented in [4], for the following reasons:
>  1. [4] is theoretically well-founded;
>  2. optimization of neural networks is nonconvex, and [4] provides results that are valid in nonconvex optimization (up to some conditions);
>  3. [4] uses the order-3 derivative of the loss, to which we have access (thanks to Section 3).
>
> The idea behind [4] (and our regularization) is that it is worth trusting the order-2 Taylor approximation of the loss as long as the cubic term of the Taylor approximation is not too large. Otherwise, it would be better to take smaller training steps.
>
> The reviewer may check the theoretical results in [4]. If the reviewer insists, we can explain why we do not use the pseudo-inverse in appendix. But, since it is (almost) never used in optimization, we believe that it is unnecessary.
>
> # References
>
> [1] *Linear and Nonlinear Programming*, Luenberger, 2008.
>
> [2] *Lectures on Convex Optimization*, Nesterov, 2018.
>
> [3] *Numerical Optimization*, Nocedal and Wright, 1999.
>
> [4] *Cubic regularization of Newton method and its global performance*, Nesterov and Polyak, 2006.
>
> [5] *Foundations of Modern Analysis*, Dieudonne, 1960.
>
> [6] *Sharp worst-case evaluation complexity bounds for arbitrary-order nonconvex optimization with inexpensive constraints*, Cartis et al., 2020.
>
> [7] *Worst-case evaluation complexity for unconstrained nonlinear optimization using high-order regularized models*, Birgin et al., 2017.

---

> ### Author Response · Authors · 2024-11-27
>
> Dear Reviewer PhVC,
>
> We would be delighted to know if you are satisfied with the changes made to our paper, including:
>  * the improvement of the quality of the writing (in the entire paper);
>  * the explanation of the notation used for the order-$d$ derivatives and tensor contraction (Section 3 and Appendix A);
>  * the answer given in our rebuttal about the pseudo-inverse.

---

> ### Comment · Reviewer_PhVC · 2024-12-01
> **Reply to the authors**
>
> Thank you for your effort and corrections made to the manuscript, I apologize for the delayed response.
> I believe that the writing improvements help clarify the results, and merit raising my rating to 5.
>
> Let me point out first of all, that my question regarding the pseudo-inverse was not the reason for my initial rating, and perhaps I did not phrase it correctly - my intension was that using SVD, even under poor conditioning ,it's possible to remove the nearly singular eigenvalues (even numerically by simply setting them to 0 in the inverse matrix) and perform the search on the subspace spanned by the non-flat (nearly flat) directions, see refs.[1-8].
>
> Secondly, the reason my score is not higher is simply that the proposed algorithm, which is a large portion (if not the main portion) of this work does not show any improvement over standard algorithms, and along with the lack of an expected convergence rate, there is no reason to believe it is particularly useful. Therefore, my main issue is that the focus of the work in its current form is not of sufficient interest to the community to merit publication in ICLR.
>
> References:
>
> [1] Golub, G.H., and Van Loan, C.F. 1989, Matrix Computations, 2nd ed. (Baltimore: Johns Hopkins
> University Press), §8.3 and Chapter 12.
>
> [2] Lawson, C.L., and Hanson, R. 1974, Solving Least Squares Problems (Englewood Cliffs, NJ:
> Prentice-Hall), Chapter 18.
>
> [3] Forsythe, G.E., Malcolm, M.A., and Moler, C.B. 1977, Computer Methods for Mathematical
> Computations (Englewood Cliffs, NJ: Prentice-Hall), Chapter 9.
>
> [4] Wilkinson, J.H., and Reinsch, C. 1971, Linear Algebra, vol. II of Handbook for Automatic Computation (New York: Springer-Verlag), Chapter I.10 by G.H. Golub and C. Reinsch.
>
> [5] Dongarra, J.J., et al. 1979, LINPACK User’s Guide (Philadelphia: S.I.A.M.), Chapter 11.
>
> [6] Smith, B.T., et al. 1976, Matrix Eigensystem Routines — EISPACK Guide, 2nd ed., vol. 6 of
> Lecture Notes in Computer Science (New York: Springer-Verlag).
>
> [7] Stoer, J., and Bulirsch, R. 1980, Introduction to Numerical Analysis (New York: Springer-Verlag),
> §6.7.
>
> [8] Golub, G.H., and Van Loan, C.F. 1989, Matrix Computations, 2nd ed. (Baltimore: Johns Hopkins
> University Press), §5.2.6. [5]

---

### Official Review · Reviewer_Hw9J · 2024-11-05

**Soundness:** 4
**Presentation:** 3
**Contribution:** 2
**Rating:** 6
**Confidence:** 4

**Summary:**

The paper proposes to make second-order optimization computationally tractable by taking a coarse view of high-dimensional parameter spaces. The authors break the Hessian of a deep learning model into blocks, with one block for every pair of layers, and compute one summary scalar per block. This allows for modeling inter-layer interactions, unlike many other approaches to approximate second-order optimization that neglect these terms. The authors propose a cubic-regularized version of their algorithm, and present experimental evidence that inter-layer interactions in deep learning models are non-trivial. Ultimately, the experimental results of the authors' proposed method are mixed.

**Strengths:**

The authors present a very interesting idea, and thoroughly motivate the idea. It really is a big short-coming that many related papers on this topic neglect inter-layer interactions when designing optimization algorithms. And finding computationally tractable ways to "summarize" the Hessian is a very strong idea.

The discussion of related work is quite comprehensive and clear.

Generally the work feels quite thoughtful: considering what are the issues with Newton's method, and how to try to overcome them.

**Weaknesses:**

I think the method could potentially be introduced in a more straightforward way, and I want to suggest one way. I think the method could be viewed as a change of variables to a smaller set of local optimization variables. In particular, instead of viewing the loss as a function of general perturbations to all the weight tensors:

loss( W_1 + ∆W_1, W_2 + ∆W_2, ..., W_S + ∆W_S)

we can view the loss as as a function of scalar-parameterized perturbations to each layer:

loss( W_1 - η_1 * G_1 , W_2 - η_2 * G_2, ..., W_S - η_S * G_S)			(*)

where η_1, η_2, ..., η_S are a collection of scalars and G_1, G_2, ..., G_S are the gradients of the loss with respect to each weight tensor, evaluated at the point W_1, W_2, ..., W_S. We can consider (*) to be a loss function with S variables η_1, η_2, ..., η_S. It's then clear that Hessian of this "reduced-dimensionality" loss is an S x S matrix. And we can throw any of a wide range of optimization methods toward solving this local S-dimensional optimization problem.

I also think the title could possibly be improved. What about something like "Multi-Tensor Optimization via Second-Order Scalar Summaries"?

Algorithmically, there is a weakness in the method that it only searches in the gradient direction for each tensor. This means the method would miss Shampoo-style [1] changes to the gradient direction which have been found to speed up deep learning training empirically.

The experimental results are mixed and perhaps not terribly promising at the moment. But it's good that you are open and up front about this. I think it's unlikely the broader community would focus on the paper too closely without more thorough experimental results.

Some light proof-reading of the writing would be helpful in places. For example line 90--91 "This is typically what is done by Dangel (2023), despite it does not go beyond the second-order derivative."

[1] https://arxiv.org/abs/1802.09568

**Questions:**

See weaknesses section. Do the authors agree with this perspective?

---

> ### Author Response · Authors · 2024-11-22
>
> We thank Reviewer Hw9J for the review and the insightful remarks and suggestions. We also appreciate that the reviewer acknowledges our motivation and the thinking behind the paper.
>
> # Quality of writing
>
> We thank the reviewer for the remark. We have carefully proof-read (again) our paper and made many minor changes (highlighted in blue). We hope that the writing quality is now acceptable.
>
> # Derivation of the optimization method
>
> Thank you for the suggestion! We fully agree with the reviewer: there exist much simpler ways to derive our optimization method, and the one proposed by the reviewer is valid. We have included it in Appendix B, along with another interpretation of our method.
>
> At start, we chose not to put it that way, because we wanted to keep the same notation in Sections 3 and 4, and our notation is more adapted to the mathematical analysis we provide in Appendix F.
>
> # Search in the gradient direction for each tensor
>
> The reviewer is right: in practice, we focus on the choice of direction $\mathbf{u} = \mathbf{g}$, and the central matrix of our computation is $\bar{\mathbf{H}}$, which is based on the Hessian.
>
> However, it is possible to choose $\mathbf{u}$ differently. For instance, one can choose $\mathbf{u} = \mathbf{A} \mathbf{g}$, where $\mathbf{A}$ is the diagonal of the inverse of the Hessian. In short, it is possible to choose $\mathbf{u}$ so that we partially take into account the curvature of the loss before computing $\bar{\mathbf{H}}$.
>
> Also, it is possible to choose differently the matrix: instead of $\bar{\mathbf{H}}$ (the Hessian projected on certain directions) we can use other matrices. For instance, one could use the "square" of the Jacobian (used in the Gauss-Newton algorithm) projected on certain directions.
>
> We do not know if there is a choice of $(\mathbf{u}, \bar{\mathbf{H}})$ that would allow us to recover Shampoo. But it is clear that tuning them would lead us to algorithms different from the one we propose here. (Besides, Shampoo does not use directly the Hessian, so, if we want to recover Shampoo, we have to replace $\bar{\mathbf{H}}$ by a matrix that only uses first-order information).
>
> # Title of the paper
>
> We thank the reviewer for the suggestion. The main difficulty is to find a title mentioning both higher-order derivatives and our optimization algorithm, while keeping it short.

---

> > ### Comment · Reviewer_Hw9J · 2024-11-26
> >
> > Thank you to the authors for responding to the feedback!
> >
> > (Please note one new grammar error in the sentence after "What are we really looking for?" in 2.3 Motivation.)

---

### Official Review · Reviewer_6Jm1 · 2024-11-08

**Soundness:** 3
**Presentation:** 2
**Contribution:** 2
**Rating:** 5
**Confidence:** 4

**Summary:**

This work considers building summaries of higher order loss derivatives, like the Hessian and the third-order Tensor, which bucket interactions at the level of layers or some arbitrary partitions, instead of each individual parameter. In particular, by considering a particular contraction (like with the all-ones vector or gradient direction), very compact higher-order summaries can be built (which scales polynomially in the number of layers, instead of parameters). As an application, they use it to derive a layerwise scaling of learning rates, which neatly interpolates between the two extremes of using Newton's method and Cauchy's steepest descent rule. The method is demonstrated on some very simple experimental setups.

**Strengths:**

- The paper is in general well motivated with the need to capture the interactions between parameters in different layers, which is often ignored by block-diagonal methods. This is operationalized in a natural way by studying the Hessian with suitable contractions.

- The approach of getting layerwise learning rates through their layerwise grouping is neat. This offers a principled extension to the Cauchy's steepest descent rule.

- The method could also find utility in studying the behaviour of feature learning across layers, and thus be used more than just in optimization.

**Weaknesses:**

- The experimental section is quite weak. I understand that the authors themselves pitch it as a proof-of-concept, but I am not so sure about even if you can call it a proof of concept. The experiments are on small datasets like CIFAR, even over there none of the methods get the typical 90% and above accuracy, the test accuracy of their method is much worse than K-FAC.

- More fundamentally, it is unclear to me where lies the bigger problem: correcting the curvature across layers, or that within the layers. It is well known that the Hessian tends to have a significant energy on its block diagonals, and thus maybe correcting across the layers, may not possess significantly more information.

- Also, even for their method, I am curious what amount of the performance can be explained by simply estimating the scales of the Hessian on the diagonal blocks. In particular, if they instead use diag(\bar{H}), and then use it to get layerwise learning rates, how does that perform? This would form a test bed to showcase how crucial is it to capture cross-layer information.

- Besides, I think in the vision setting the Hessian tends to be more homogeneous across the networks as opposed to that in Transformers with language modelling [1]. Hence, I think their approach might be more suited to that setting, and would be interesting to see if it can outperform methods like Adam-Mini [2].

- There are very limited baselines considered by the authors. I would have liked to see the Cauchy step size, AdaHessian, and even a block-diagonal quasi Newton method.

- The overall runtime cost can be quite large, as there are multiple Hessian vector products. Can the authors do a wall-clock comparison?

References:

[1] Ormaniec, et. al. (2024). What Does It Mean to Be a Transformer? Insights from a Theoretical Hessian Analysis. arXiv preprint arXiv:2410.10986.

[2] Zhang, et. al. (2024). Adam-mini: Use fewer learning rates to gain more. arXiv preprint arXiv:2406.16793.

**Questions:**

See above

---

> ### Author Response · Authors · 2024-11-22
>
> We thank Reviewer 6Jm1 for the review and the insightful remarks and questions. We also appreciate that the reviewer acknowledges our motivation and the relevance of our method (with potential applications beyond optimization).
>
> # Do we need cross-layer information?
>
> > if they instead use diag(\bar{H}), and then use it to get layerwise learning rates, how does that perform? This would form a test bed to showcase how crucial is it to capture cross-layer information.
>
> We provide such experiments in **Appendix K**.
>
> This is indeed an crucial question: do we really need cross-layer information? As mentioned at the end of Section 5, we provide such a study in Appendix K. We have tested our method (using full $\bar{\mathbf{H}}$) against its diagonal version (using only the diagonal of $\bar{\mathbf{H}}$) with LeNet and VGG11'. As the reviewer will notice (see Fig. 6), keeping the off-diagonal coefficients of $\bar{\mathbf{H}}$ leads to better and more stable optimization results. Please note that for a fair comparison with our method, we have tested a wide range of hyperparameters for the diagonal version.
>
> Also, we believe that the reviewer would be interested in the results presented in **Appendix J**. We show how the choice of the partition affects training (we have tested several choices of partition of VGG11').
>
> # Inter-layer or intra-layer correction of the curvature?
>
> > it is unclear to me where lies the bigger problem: correcting the curvature across layers, or that within the layers.
>
> According to the experiments performed in Appendix K (see previous answer), it is important to consider inter-layer interactions. The combination of inter-layer and intra-layer interactions has not yet been fully achieved.
>
> But, in theory, our method is flexible enough to incorporate part of the intra-layer curvature. Since it is possible to choose any custom candidate descent direction $-\mathbf{u}\_t$, one may choose a $\mathbf{u}\_t$ different from the gradient $\mathbf{g}\_t$, such as $\mathbf{u}\_t = \mathbf{A} \mathbf{g}_t$, where $\mathbf{A}$ is the diagonal of the inverse of the Hessian. In short, it is possible to choose $\mathbf{u}\_t$ so that we partially take into account the curvature of the loss before computing $\bar{\mathbf{H}}$.
>
> # Test accuracy
>
> This is a tough question.
>
> We have decided to focus our experimental evaluation on the optimization process. Thus, we focus on the *training loss* metric and we did not use any of the common regularization techniques (data augmentation, weight decay/penalty, drop-out, SAM, etc.). This may explain the accuracies obtained on the test set.
>
> Here are the reasons of our choice:
>  1. the proposed training algorithm is designed as an optimization method (just like the SGD, Newton's method, etc.), and not as a procedure to improve generalization (see Appendix B for the initial derivation of the method). Given that context, we have considered that the most relevant metric is the *training loss* (and not the training accuracy or any metric related to the test set);
>  2. we know that some optimization methods, such as the SGD, produce good generalization results, even though they were originally designed for optimization. However, this "implicit regularization" phenomenon is not yet fully understood in the case of deep non-linear neural networks. To avoid reporting a metric related to a poorly explained phenomenon, we preferred to stick with the training loss.
>
> If a user is interested in the test loss, we think that our method can be used just like other optimizers with data augmentation, weight decay, etc.
>
> But we are aware that the "gold standard" in classification is the test accuracy, since that is the metric that we really looking to optimize. This is why we provide test accuracies (Appendix I).
>
> **Please note that Figure 5 (in Appendix I) illustrates very well that comparing test losses/accuracies can be tricky.** For instance, the test losses and accuracies of VGG11' exhibit different behaviors: at several epochs (see epoch 15), best-performing models are different when considering loss and accuracy; and an exploding test loss may not imply a drop in accuracy. **So, some uncontrolled phenomena are at work. This illustrates the reasons why we focus on the training loss.**
>
> However, we would understand if the reviewer does not share this view.
>
> # Wall-time comparison
>
> The reviewer is perfectly right, such a comparison was missing in the initial version. We have put it in Appendix M in the current version.
>
> # Application to transformers
>
> We thank the reviewer for the suggestion! We hope we will be able to implement it and test it soon.

---

> ### Author Response · Authors · 2024-11-27
>
> Dear Reviewer 6Jm1,
>
> We would be delighted to know if you are satisfied with our answer and the changes made to our paper, including:
>  * testing $\bar{\mathbf{H}}$ against $\mathrm{diag}(\bar{\mathbf{H}})$ with our method (see Appendix K); that is, testing the usefulness of off-diagonal coefficients of $\bar{\mathbf{H}}$, that we are able to calculate thanks to our method;
>  * our answer about the relevance of the test accuracy in our setup.

---

### Author Response · Authors · 2024-11-22

Once again, we thank all reviewers for their comments, suggestions, and questions. We are pleased that all reviewers showed interest, if not enthusiasm, for our work and made relevant suggestions for improvement.

Several sections have been added and placed at the end of the appendix to maintain the current section numbering, but we plan to reorder these in future revisions of the paper.

# Quality of writing

We take the comments of Reviewers Hw9J and PhVC about the quality of writing seriously. Therefore, we have performed many corrections in the main text (highlighted in blue). We hope that the writing quality of the paper is now sufficient.

# Comparison of training times

It is true that a comparison of training times is missing, as pointed out by Reviewers 6Jm1 and PhVC. Without such a comparison, one could legitimately think that our method is very expensive.

To fix this, we have added Appendix M and a table comparing the training times: the training with our method is not excessively longer than with K-FAC, and is of the same order as Adam/K-FAC on small networks.

# Appendices J and K

If possible, we encourage the reviewers to take a look at Appendices J and K. Appendix K contains a study showing the importance of keeping off-diagonal coefficients of $\bar{\mathbf{H}}$, which illustrates the value of our method. Appendix J shows the influence of the choice of the partition of the parameter space on the training time and the final loss.

# Notations in Section 3

Reviewer PhVC pointed out that the explanation of the notations used in Section 3 was not sufficient. We agree with this view, and we have worked hard to make this as clear as possible in the new versions of the paper:
 1. several paragraphs of Section 3 have been rewritten;
 2. Appendix A has been improved;
 3. some notations have been improved;
 4. Appendix L has been added to explain thoroughly our notation.

However, due to space limitations, we have not included these explanations in the main text. Otherwise, we would have to remove essential parts of the paper.

# Interpretations of the non-regularized optimization method

Following the suggestion of Reviewer Hw9J, we have added two paragraphs in Appendix B showing how to derive our method in a simpler way.

---

### Author Response · Authors · 2024-11-28

To provide a complete answer to Reviewer PhVC, we have added a paragraph in Appendix A, explaining thoroughly the implementation of the method presented in Section 3.

We hope that it is now easier to understand how it is possible to compute efficiently the tensors $\mathbf{D}_{\boldsymbol{\theta}}^d(\mathbf{u})$ containing order-$d$ information about the loss.

Moreover, we have added some details about the symmetric structure of  $\mathbf{D}_{\boldsymbol{\theta}}^d(\mathbf{u})$, which allows us to compute it even more efficiently: we compute and store only $\frac{(S+d-1)!}{d! (S-1)!}$ coefficients instead of $S^d$. For example, with $S = 10$ groups of parameters and $d = 3$ (derivative of order 3), we store only $220$ coefficients instead of $S^d = 1000$.

---

### Author Response · Authors · 2024-12-03

We thank again the reviewers for their time and the suggestions that helped us to improve our paper.

Alongside the main strengths of paper, which are acknowledged by all the reviewers (well-principled way of computing per-layer learning rates, measuring interactions between layers, efficient computation of quantities related to higher-order derivatives...), we are aware that the empirical evaluation of our **optimization method** is limited. However, the provided empirical evaluation goes beyond synthetic datasets and small neural networks (we even show how to use our method with a 100-layers MLP), which we call a "proof-of-concept".

Overall, we think that the **methodological tools** developed in our paper are of interest to the community (as acknowledged by the reviewers), and we have built upon them an optimization method that is applicable in several (non-trivial) settings. This optimization method, although not thoroughly evaluated, helps to show how the tools we developed can be used, and provides practical insights for future research.

---

### Meta-Review · Area_Chair_wTAM · 2024-12-17

**Metareview:**

This paper proposes an optimization algorithm “NewtonSummary”, which can be viewed as a middle ground of Newton’s method and Cauchy’s steepest descent. By grouping parameters and calculating “summarized” high-order derivatives of the loss function, the algorithm implements a computationally inexpensive (but crude) version of Newton’s method with Nesterov cubic regularization.

The reviewers agreed that the algorithm has some novel elements such as a computationally tractable way to summarize 2nd and 3rd order derivatives of the loss function. Indeed, as pointed out by one of the reviewers during reviewer-AC discussion, the computational method for higher order information presented in the paper can be useful, in particular for interpretability and understanding the landscape of DNNs.

However, it was unanimously pointed out that the empirical demonstration of the proposed algorithm’s effectiveness leaves much to be desired. Although the authors claims to provide a preliminary “proof-of-concept”, the algorithm is evaluated in a limited range of benchmarks and models (although the authors indeed test it on a 100-layer model), is compared against only a few baselines, and yet does not show clear advantages over existing methods. This makes me question the motivation for using higher-order methods for neural network training—if gradient-based methods like adam are better, why use this method? Moreover, the paper does not provide any convergence guarantees, which is also a weakness.

Overall, my conclusion is that the paper is slightly below the acceptance threshold for ICLR.

**Additional Comments On Reviewer Discussion:**

The summary includes a summary on the discussion of strengths/weaknesses raised by the reviewers. Additionally, Rev Hw9J pointed out that the method could be described in an alternative form and the authors reflected that in Appendix B.

---

### Decision · Program_Chairs · 2025-01-22

Reject